# A SARS-CoV-2 variant-adjusted threshold of protection model for monoclonal antibody pre-exposure prophylaxis against COVID-19

Rhiannon Edge[1,10], Sam Matthews [1,10], Bahar Ahani[2], Anastasia A. Aksyuk [3], Lindsay Clegg[4], John L. Perez[5], Mark T. Esser[5], Lee-Jah Chang[5], Ian Hirsch[1], Tonya Villafana[5], John Pura[6], Oleg Stepanov[7], Katie Streicher[3], Tom White[1], Taylor S. Cohen[5], Dean Follmann [8], Peter B. Gilbert [9] & Seth Seegobin[1] ✉

Clinical development of monoclonal antibodies (mAbs) against severe acute respiratory syndrome coronavirus 2 (SARS-CoV-2) is challenging due to rapid changes in the variant landscape. This study identified a threshold model for neutralising antibody (nAb) titres associated with clinically relevant protection against symptomatic COVID-19 for vulnerable populations. Using efficacy data from the phase 3 PROVENT pre-exposure prophylaxis trial of tixagevimab–cilgavimab (NCT04625725), individual nAb $ID_{50}$ titres were predicted by dividing serum mAb concentration by prevalence-adjusted tixagevimab–cilgavimab potency (from in vitro $IC_{50}$ values combined with viral surveillance data) and related to efficacy with a Cox model. The Threshold of Protection (ToP) Cox model was externally validated using data from the phase 3 SUPERNOVA trial (NCT05648110), which assessed sipavibart efficacy against symptomatic COVID-19 in immunocompromised participants. The PROVENT ToP model estimated the variant-specific observed efficacies from SUPERNOVA for 3 and 6 months post any dose with Lin's concordance of 0.86 and 0.75, respectively. This approach integrates predicted nAb $ID_{50}$ titres against multiple SARS-CoV-2 variants into a ToP model that can be applied across different variants and could serve as a surrogate endpoint in immunobridging studies to expedite clinical evaluation and regulatory approval for mAbs targeting SARS-CoV-2.

Immunocompromised individuals with suboptimal vaccination responses are at increased risk of severe coronavirus disease 2019 (COVID-19) outcomes compared with the general population[1,2]. Monoclonal antibodies (mAbs) demonstrated effectiveness for COVID-19 prevention during the pandemic, including in immunocompromised populations[3–5]. Tixagevimab–cilgavimab (previously AZD7442) are severe acute respiratory syndrome coronavirus 2 (SARS-CoV-2)–neutralising mAbs including YTE and TM amino acid modifications

[1]Biometrics, Vaccines & Immune Therapies, BioPharmaceuticals R&D, AstraZeneca, Cambridge, UK. [2]Bioinformatics, Vaccines & Immune Therapies, BioPharmaceuticals R&D, AstraZeneca, Gaithersburg, MD, USA. [3]Translational Medicine, Vaccines & Immune Therapies, BioPharmaceuticals R&D, AstraZeneca, Gaithersburg, MD, USA. [4]Clinical Pharmacology and Quantitative Pharmacology, Clinical Pharmacology & Safety Sciences, R&D, AstraZeneca, Gaithersburg, MD, USA. [5]Vaccines & Immune Therapies, BioPharmaceuticals R&D, AstraZeneca, Gaithersburg, MD, USA. [6]Biometrics, Vaccines & Immune Therapies, BioPharmaceuticals R&D, AstraZeneca, Gaithersburg, MD, USA. [7]Clinical Pharmacology and Quantitative Pharmacology, Clinical Pharmacology & Safety Sciences, R&D, AstraZeneca, Cambridge, UK. [8]Biostatistics Research Branch, National Institute of Allergy and Infectious Diseases, National Institutes of Health, Bethesda, MD, USA. [9]Vaccine and Infectious Disease Division, Fred Hutchinson Cancer Center, Seattle, WA, USA. [10]These authors contributed equally: Rhiannon Edge, Sam Matthews. ✉e-mail: Seth.Seegobin@astrazeneca.com

for extending mAb half-life and reducing theoretical risk of antibody-enhanced disease[6]. Tixagevimab–cilgavimab demonstrated efficacy for pre-exposure prophylaxis of COVID-19 in the PROVENT clinical trial, conducted in a population of high-risk but mainly non-immunocompromised participants[7]. However, tixagevimab–cilgavimab did not show neutralising activity against Omicron subvariants including BQ.1.1 and XBB[8,9]. Following tixagevimab–cilgavimab, sipavibart (previously AZD3152), another YTE- and TM-modified mAb, demonstrated efficacy for prevention of COVID-19 in an immunocompromised population in the SUPERNOVA clinical trial, at a time when susceptible variants circulated[10]. However, efficacy was not shown against resistant variants that emerged during the trial[10]. This highlights the challenges in developing new SARS-Cov-2–neutralising mAbs targeting contemporary SARS-CoV-2 variants. An alternative approach is needed to accelerate the development of neutralising mAbs amidst a rapidly changing variant landscape.

Neutralising antibody (nAb) titres are a variant-specific measure of the neutralising potency of serum samples[11]. In March 2024, the United States Food and Drug Administration issued an Emergency Use Authorization for pemivibart based on an immunobridging approach involving comparison of the nAb titre of pemivibart 4500 mg against the contemporary JN.1 variant with the nAb titre previously determined for adintrevimab 300 mg against the historic Delta variant[12]. The methodology was reliant on direct bridging to the observed historical efficacy estimate of 71%. An issue with immunobridging to historic trials is that the binary null hypothesis is strictly linked to observed efficacy in the historic trial without flexibility. The ability to identify nAb-titre targets associated with different efficacy levels would facilitate evaluation of neutralising mAbs against emerging SARS-COV-2 variants. Moreover, direct bridging must be done by variant and therefore fails to characterise protection against a complex variant landscape as currently seen. Prior analyses have not attempted to combine data against multiple variants or adjust for changing viral variant prevalence[13]. Thus, other approaches are needed that can account for a mAb having different neutralisation activities against multiple variants when estimating COVID-19 efficacy.

Correlates of protection are immunological markers that can be used to reliably predict efficacy against clinically relevant endpoints and have been used for influenza and COVID-19 vaccine licensing[14–17]. For neutralising mAbs, an analogous serum mAb concentration above which clinically relevant protection against specific viruses is expected, or threshold of protection (ToP), would have potential to help facilitate clinical development. The risk of developing symptomatic COVID-19, and COVID-19 severity, inversely correlates with SARS-CoV-2 nAb titres in serum, suggesting nAb titres could be used to develop a ToP for COVID-19 mAbs[18–21]. It is reasonable to assume that nAb titres are directly related to the ability of a mAb to protect from infection, especially for mAbs with Fc modifications limiting their activity solely to neutralisation. Serum mAb concentrations combined with variant-specific in vitro 50% inhibitory concentration ($IC_{50}$) values that reflect mAb potency, can predict nAb 50% inhibitory dilution ($ID_{50}$) titres across multiple SARS-CoV-2 variants[21].

Data from PROVENT and SUPERNOVA provide an opportunity to quantify the relationship between potency-scaled serum mAb concentrations for a range of SARS-CoV-2 variants[21] and risk of symptomatic COVID-19, facilitating development of ToPs for COVID-19 mAbs. We explored applicability of a ToP model based on tixagevimab–cilgavimab data from PROVENT, conducted in a largely non-immunocompromised population early in the pandemic largely dominated by a small number of variants including Alpha, Delta, and early Omicron subvariants[7,22], to the sipavibart data from SUPERNOVA, conducted in an immunocompromised population with a varied SARS-CoV-2 variant landscape involving multiple Omicron subvariants[10]. Here, we describe a methodological approach for determining a ToP for mAbs targeting SARS-CoV-2 associated with clinically relevant protection against symptomatic COVID-19 (Fig. 1).

## Results

### Prevalence-adjusted daily nAb $ID_{50}$ titres

The SARS-CoV-2 variant landscape during PROVENT was characterised by three distinct waves of dominant variants, Alpha ($IC_{50}$: 2.1 ng/mL), Delta ($IC_{50}$: 2.2 ng/mL), and the Omicron BA.1 ($IC_{50}$: 171.1 ng/mL) and BA.1.1 ($IC_{50}$: 466 ng/mL) waves that began towards the end of the study period (Supplementary Fig. S1). The variant landscape during SUPERNOVA was more complex, including a mixture of JN.1 and SARS-CoV-2 variants with F456X or L455X mutations, with sipavibart potency varying notably across these lineages (Supplementary Fig. S2). Figure 2 presents the prevalence-adjusted $IC_{50}$ for both studies (panel A), and serum mAb concentration over time (panel B). Serum mAb concentration was divided by the prevalence-adjusted $IC_{50}$ to produce the prevalence-adjusted nAb $ID_{50}$ titres over time (panel C). For PROVENT, during the periods dominated by the Alpha and Delta variants, against which tixagevimab–cilgavimab showed potent neutralising activity ($IC_{50}$ <5 ng/mL), the prevalence-adjusted nAb $ID_{50}$ titre profile diminished at a constant rate (on a log scale), consistent with natural elimination of the mAb in serum over time. At the onset of the BA.1/BA.1.1 period, titres declined rapidly due to decrease of tixagevimab–cilgavimab neutralising activity against these variants. The prevalence-adjusted nAb $ID_{50}$ titre profile for SUPERNOVA was lower and more consistent over time, due to minimal fluctuations in prevalence-adjusted $IC_{50}$ values and redosing of sipavibart after 6 months.

### PROVENT ToP model

Efficacy against symptomatic COVID-19 was estimated using the ToP model. Over 1 year of follow-up in PROVENT, there were 63 events amongst 3272 participants (1.8%) in the tixagevimab–cilgavimab group with available serum mAb concentration predictions from the population pharmacokinetic (popPK) model, and 55 events amongst 1730 participants (3.2%) in the placebo group[22]. The relationship between prevalence-adjusted nAb $ID_{50}$ titre at time of exposure and efficacy against symptomatic SARS-CoV-2 infection based on PROVENT study data only is shown in Fig. 3. As different applications may require different considerations of a minimal clinical meaningful level of protection, several threshold values are presented in Table 1.

The instantaneous efficacy estimated from the PROVENT ToP model was compared with that observed in SUPERNOVA (Fig. 4). There was good agreement between model-predicted and observed instantaneous overall efficacy following a single dose of the investigational product. In accordance with standard endpoints in clinical trials, a comparison between PROVENT ToP model-predicted average overall efficacy and observed average overall efficacy from SUPERNOVA was undertaken. Figure 5 shows good agreement with a mean absolute difference of 5.0%. As data and events accrued over time, variability in the Cox proportional-hazards model–derived average overall efficacy decreased, resulting in convergence between the observed efficacy in SUPERNOVA and the predictions from the PROVENT ToP model. The 95% confidence intervals (CIs) from the PROVENT ToP model were conservative compared with those estimated from observed data.

Figure 6 provides PROVENT ToP model predictions corresponding to endpoints in SUPERNOVA[10]. Model predictions were well aligned with observed variant-specific efficacy results across pre-specified endpoints from this study for 3 and 6 months post any dose with Lin's concordance correlation coefficient (CCC) (95% CI) of 0.86 (0.45, 0.89) and 0.75 (0.28, 0.92), respectively (Fig. 7).

### PROVENT + SUPERNOVA ToP model

The PROVENT + SUPERNOVA ToP model analysis additionally included 122 events amongst 1649 participants (7.4%) in the sipavibart

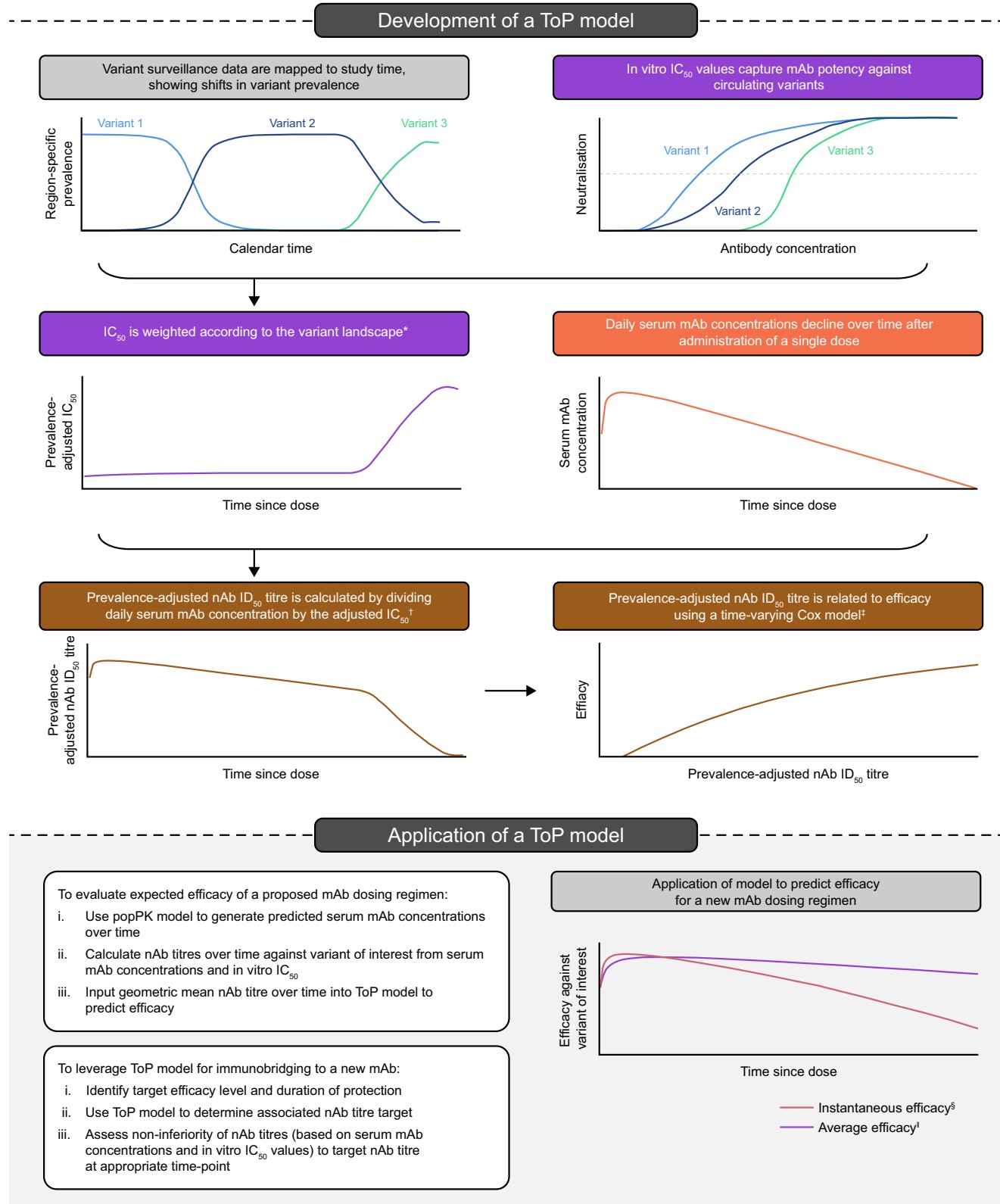

**Fig. 1 | Development and application of a ToP model.** For further details of the ToP model application see the Supplementary Methods. *Prevalence-adjusted IC$_{50}$ increases as a mAb becomes less effective against dominant variants. †Prevalence-adjusted nAb ID$_{50}$ titre is a function of decreasing serum mAb concentration and inverse mAb potency. ‡Efficacy is calculated as 100(1 − hazard ratio) (%) with respect to the treatment-naive group. §Instantaneous efficacy predictions are generated by evaluating the ToP model at the geometric average prevalence-adjusted nAb ID$_{50}$ titre at each study day. These provide a snapshot of the current efficacy at the given timepoint. ‖Average efficacy predictions are generated by evaluating the ToP model at the geometric average prevalence-adjusted nAb ID$_{50}$ titre up to each study day. These provide an assessment of the efficacy post-dose through to the given time-point. IC$_{50}$ 50% inhibitory concentration, ID$_{50}$ 50% inhibitory dilution, mAb monoclonal antibody, nAb neutralising antibody, popPK population pharmacokinetic, ToP threshold of protection.

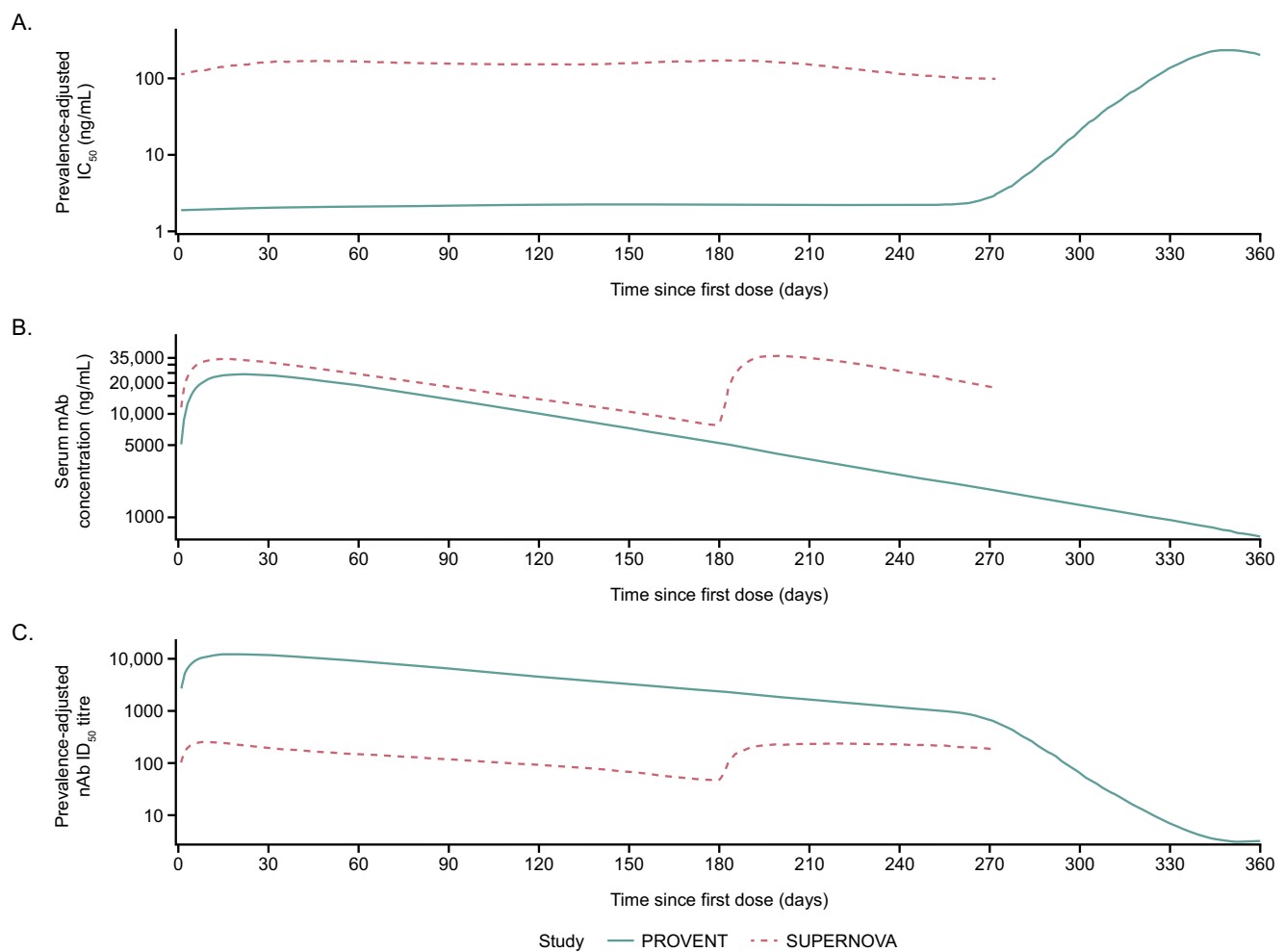

**Fig. 2 | Geometric mean of prevalence-adjusted IC$_{50}$ (ng/mL), predicted serum mAb concentration (ng/mL), and prevalence-adjusted nAb ID$_{50}$ titres over time by study. A** Counts of sequenced SARS-CoV-2 variant data from GISAID were combined with the individual-level follow-up times of risk to derive participant-level on-study prevalence for each timepoint. The geometric mean of these over time since first dose is presented here. **B** The geometric mean–predicted serum mAb concentration over time since first dose is presented here. **C** Prevalence-adjusted nAb ID$_{50}$ titres were derived for each participant and timepoint as serum mAb concentration (ng/mL)/prevalence-adjusted IC$_{50}$ (ng/mL). The geometric mean of these over time since first dose is presented here. Given the event-driven data cutoff in SUPERNOVA causing differential participant follow-up, data is presented up to the minimum of the last event time in the two arms. GISAID Global Initiative on Sharing All Influenza Data, IC$_{50}$ 50% inhibitory concentration, ID$_{50}$ 50% inhibitory dilution, mAb monoclonal antibody, nAb neutralising antibody, SARS-CoV-2 severe acute respiratory syndrome coronavirus 2.

group with available serum mAb concentration predictions, and 178 events amongst 1631 participants (10.9%) in the comparator group[10].

Figure 8 presents the PROVENT + SUPERNOVA ToP model. Parameter estimates can be found in Supplementary Table S1. The range of prevalence-adjusted nAb ID$_{50}$ titres observed in SUPERNOVA was much narrower (and centrally located in the overall distribution) than in PROVENT (Supplementary Fig. S3). Increased precision around the mid-point of the ToP model curve could be achieved with the inclusion of both studies, though a small divergence between the two model curves at the extreme ends was noted.

## Discussion

Although COVID-19 is no longer considered a global health emergency, SARS-CoV-2 will continue to evolve and new variants will emerge, presenting an ongoing threat to immunocompromised groups[1,2]. Therefore, rapid response to new SARS-CoV-2 variants through development of new vaccines, therapeutics, or mAbs remains a priority. An appropriate mAb ToP used as a surrogate endpoint for assessing drug efficacy against COVID-19, without the need for large efficacy trials, would advance our ability to provide

immunocompromised individuals with timely and clinically relevant protection against COVID-19.

Based on our ToP analysis, we determined that a nAb ID$_{50}$ titre value of 195 corresponds to 40% efficacy against symptomatic COVID-19 in a pre-exposure prophylaxis setting. This titre value is based on a methodological approach to deriving a ToP model for COVID-19 mAbs and was validated using SUPERNOVA data. The PROVENT ToP model shows good agreement with observed data from the SUPERNOVA analysis[10], supporting extrapolation to new mAbs and participants with immunocompromising conditions.

Predicted nAb ID$_{50}$ titres were derived as previously described[21] by dividing serum mAb concentration by IC$_{50}$. By directly relating efficacy to nAb ID$_{50}$ titres based on serum mAb concentrations, this approach is considered conservative as it ignores the effect of baseline nAb titres due to previous SARS-CoV-2 vaccinations or infections. This approach assumes a Hill coefficient of 1 and was shown to give reasonable concordance (Supplementary Fig. S4). Without this assumption, the generation of observed nAb ID$_{50}$ titre data from patient sera for each emergent variant would be required to estimate the relationship between serum mAb concentrations and nAb levels before this model could be applied. As this work aimed to rapidly make

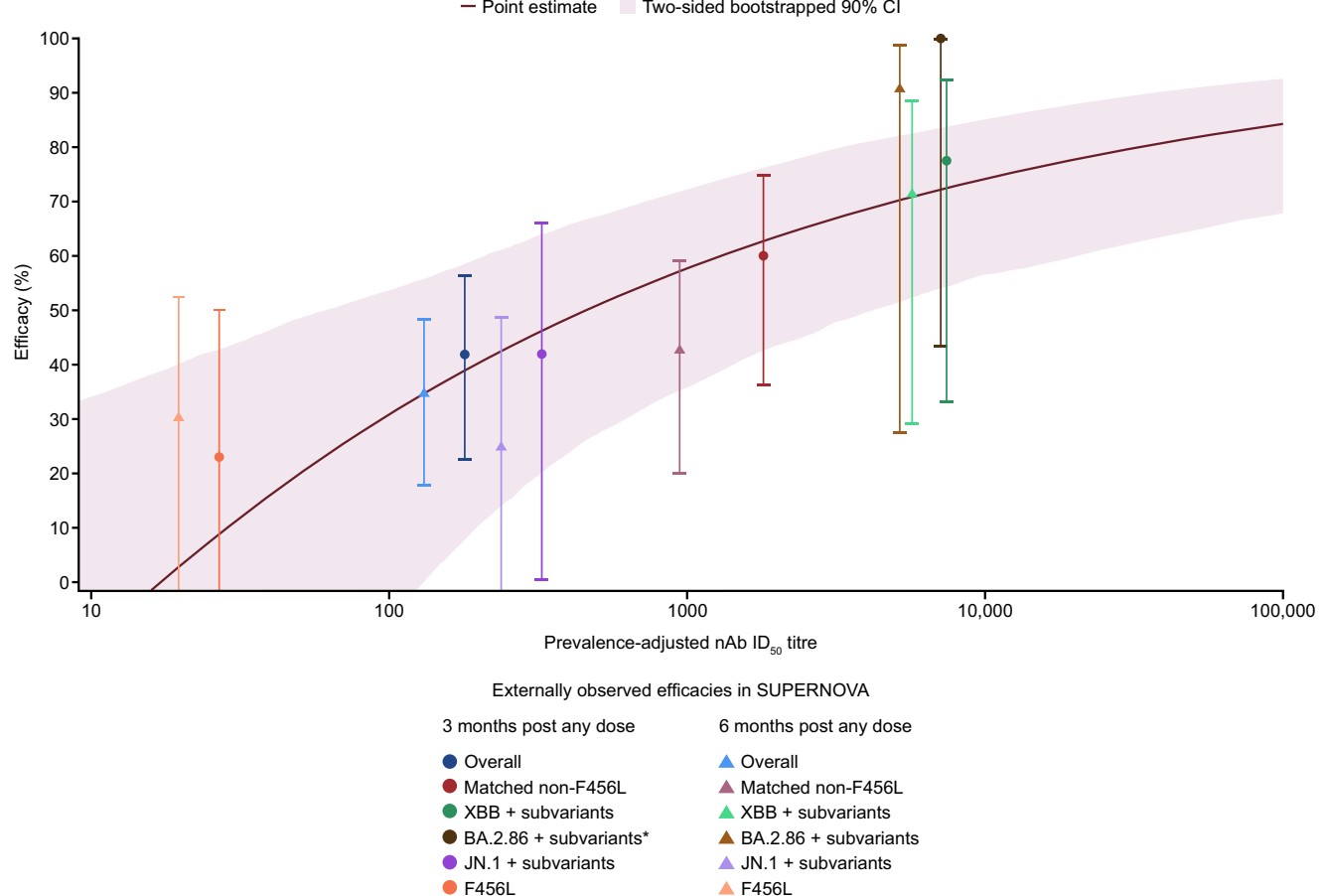

Externally observed efficacies in SUPERNOVA

**3 months post any dose**
- Overall
- Matched non-F456L
- XBB + subvariants
- BA.2.86 + subvariants*
- JN.1 + subvariants
- F456L

**6 months post any dose**
- Overall
- Matched non-F456L
- XBB + subvariants
- BA.2.86 + subvariants
- JN.1 + subvariants
- F456L

**Fig. 3 | PROVENT ToP analysis of efficacy (%) against RT-PCR–confirmed symptomatic COVID-19 through day 366 by prevalence-adjusted titre compared with observed efficacy from SUPERNOVA.** A time-varying Cox model was fit using PROVENT data, adjusting for treatment and its interaction with $\log_{10}$(prevalence-adjusted nAb $ID_{50}$ titres + 1). Efficacy was calculated as 100(1 − hazard ratio) (%) (solid purple line). The two-sided 90% CI (shaded purple region) for the point estimate of efficacy was bootstrapped using 1000 resamples with replacement. The SUPERNOVA observed efficacies were defined based on planned intervention as 100(1 − relative risk) (%) of sipavibart versus comparator, where relative risk and two-sided 95% CI was evaluated with Poisson regression with robust variance, which includes treatment and randomisation stratification factors as covariates and adjusts follow-up time. Randomisation stratification factors were COVID-19 vaccination status within 6 months before randomisation (Yes or No), SARS-CoV-2 infection within 6 months before randomisation (Yes or No), and tixagevimab–cilgavimab use within 12 months before randomisation (Yes or No). When events are not present in at least one arm, an exact approach is followed and the 97.5% two-sided CI presented[10]. SUPERNOVA observed efficacy estimates are plotted based on the average nAb $ID_{50}$ titres over 90 days or 180 days post any dose according to the timepoint. For the overall endpoint the average prevalence-adjusted nAb $ID_{50}$ titres were considered after mapping GISAID prevalence data to at-risk participants for all variants with ≥5% prevalence. For the matched non-F456L and XBB+ subvariant endpoints, a similar approach was followed with the exclusion of F456X variants (which are otherwise assumed an $IC_{50}$ of 1000), and the inclusion of only XBB subvariants based on hedgehog variants that included a XBB Pango lineage, respectively. For other analyses, daily serum mAbs estimated from population pharmacokinetics were used to predict nAb $ID_{50}$ titres in at-risk participants based on $IC_{50}$ values of 3.8, 83.1, and 1000 ng/mL for BA.2.86, JN.1 subvariant analyses and variants with F456L mutations, respectively. CI confidence interval, COVID-19 coronavirus disease 2019, GISAID Global Initiative on Sharing All Influenza Data, $IC_{50}$ 50% inhibitory concentration, $ID_{50}$ 50% inhibitory dilution, mAb monoclonal antibody, nAb neutralising antibody, RT-PCR reverse transcription-polymerase chain reaction, SARS-CoV-2 severe acute respiratory syndrome coronavirus 2, ToP threshold of protection.

predictions of mAb efficacy against emerging or future variants, the assumption of a Hill coefficient of 1 was made to create a practical and fit-for-purpose framework[21].

Several models were explored in the development of the PROVENT ToP model (Supplementary Methods). Parameter estimates are reported in Supplementary Table S2, except for the nAb $ID_{50}$ titre intercept model. This fit based on 5-fold cross validation to optimise the nAb $ID_{50}$ titre intercept and is reported in Supplementary Fig. S5. Threshold curves from all models are overlayed in Supplementary Fig. S6. The efficacy intercept model is proposed. This model suggests negative efficacy values at low nAb titres, which was not observed. However, we believe that allowing greater model flexibility is advantageous compared with the no intercept model, particularly as the CIs are otherwise fixed as a function of the single covariance parameter which yields a misleading high degree of confidence (Supplementary

Fig. S6). When the no intercept model was used to externally assess instantaneous and average overall efficacy during SUPERNOVA, the lack of fit was immediately obvious (Supplementary Fig. S7), especially when compared with Figs. 4 and 5. Similarly, the no intercept model resulted in worse concordance (Supplementary Fig. S8) when compared with Fig. 7. As well as lack of fit, the results from the no intercept model are anti-conservative, for example this would infer that a nAb $ID_{50}$ titre value of 41 instead of 195 corresponds to 40% efficacy against symptomatic COVID-19. This illustrates the importance of the model flexibility gained from the efficacy intercept. The threshold curve was robust to the inclusion of baseline covariates (Supplementary Fig. S6) so this further adjustment was not proposed. The nAb $ID_{50}$ titre intercept overestimates efficacy with worse concordance when compared with the no intercept model (Supplementary Fig. S8) and the efficacy intercept model (Fig. 7). For the proposed efficacy intercept

model, two transformations of the prevalence-adjusted nAb $ID_{50}$ titres were explored (Supplementary Table S3) and the transformation $\log_{10}$ (prevalence-adjusted nAb $ID_{50}$ titre + 1) was found to result in the smallest Akaike Information Criterion. These results help to build confidence in the chosen ToP model. The PROVENT + SUPERNOVA ToP model had a similar parameterisation to the PROVENT ToP model and was well aligned at the mid-point of the nAb-efficacy curve (Fig. 8).

### Table 1 | ToP values corresponding to varying levels of RT-PCR–confirmed symptomatic COVID-19 protection for the PROVENT ToP model

| Efficacy threshold | Required nAb GMT | Converted to IU/mL[a] |
|---|---|---|
| 40% | 195 | 28 |
| 50% | 458 | 65 |
| 60% | 1297 | 185 |
| 70% | 4959 | 708 |

A time-varying Cox model was fit adjusting for treatment and its interaction with the prevalence-adjusted nAb $ID_{50}$ titres. Efficacy was calculated as 100(1 – hazard ratio) (%). The nAb $ID_{50}$ titres required to achieve the specified efficacy levels are presented here.

COVID-19 coronavirus disease 2019, GMT geometric mean titre, $ID_{50}$ 50% inhibitory dilution, IU international units, nAb neutralising antibody, RT-PCR reverse transcription-polymerase chain reaction, ToP threshold of protection.

[a]Required nAb $ID_{50}$ titres converted to IU/mL based on a conversion factor of 0.1428, which was developed using the Wuhan D614 variant.

There is now substantial evidence showing that serum nAbs are a major protective mechanism against COVID-19[23]. In contrast to vaccines, and other long-acting mAbs which retain Fc effector function, the mechanism of action of mAbs such as tixagevimab–cilgavimab are directly related to the ability of the mAb to neutralise the virus. Despite this, the scientific community is yet to agree on a suitable ToP endpoint corresponding to efficacy that could be used by drug manufacturers and regulators to rapidly evaluate new mAbs targeting COVID-19. Key barriers include a limited ability to facilitate direct comparisons between different assays, and a lack of agreed methodology to establish a target threshold of clinically relevant protection against which new mAbs could be evaluated. Previous work shares methodological similarities with the approach presented here and showed that predictive nAb titres correlated with protective mAb efficacy, although with more uncertainty at lower titres due to a low number of events, and without adjustment for different variants in circulation[24].

The methodology presented here used surveillance data capturing variant prevalence during the clinical trials[7,22]. Previous methods have divided participants into subset analyses to account for changes in the variant landscape based on enrolment times[25]. The approach described here allows for inclusion of all variants in a single model, without exclusion of study participants. It also allows for accounting for multiple SARS-Cov-2 variants circulating at the same time. An important strength of the work presented here is that the PROVENT data spanned a period in which multiple SARS-CoV-2 variants were in

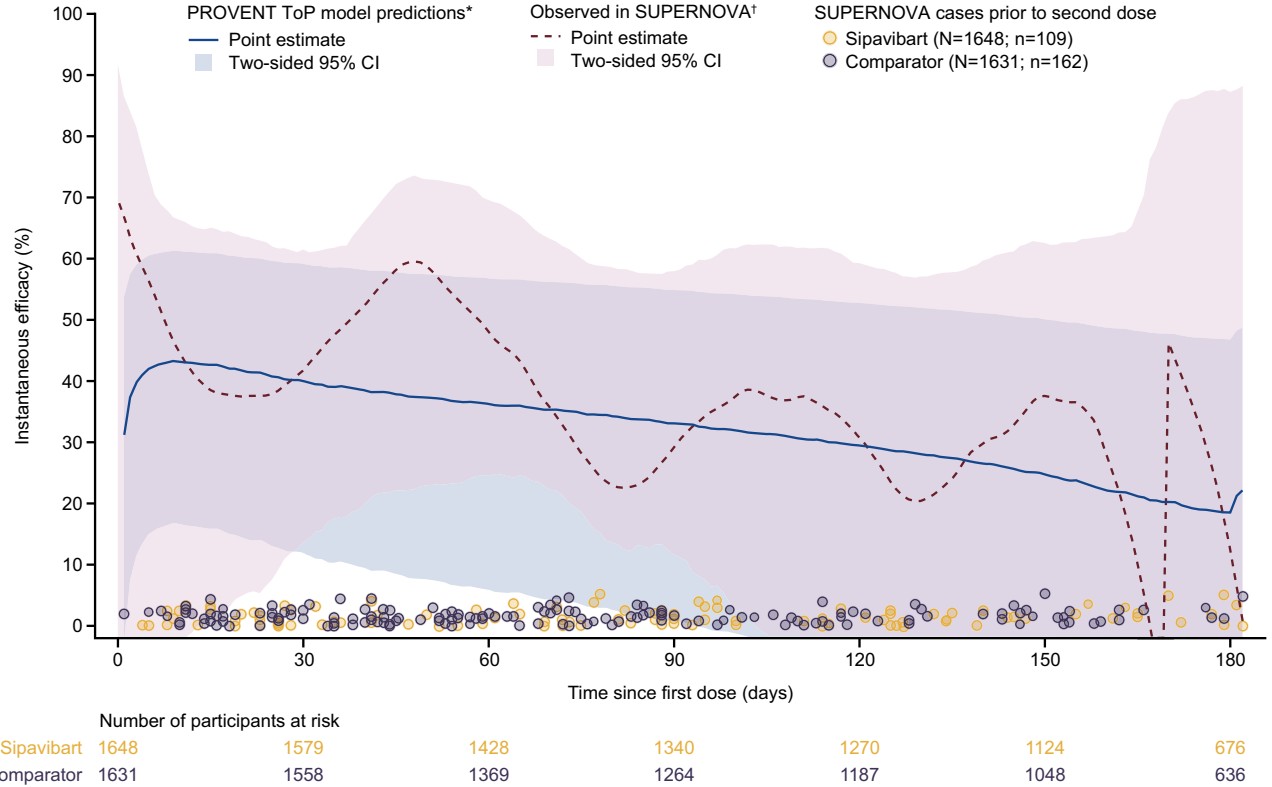

**Fig. 4 | External assessment of instantaneous overall efficacy from PROVENT ToP model using SUPERNOVA data.** *The ToP model was developed against PROVENT data using a time-varying Cox model with adjustment for treatment and its interaction with $\log_{10}$(prevalence-adjusted nAb $ID_{50}$ titres + 1). The ToP model is used to estimate instantaneous overall efficacy (solid blue line) and two-sided 95% CI (shaded blue region) from SUPERNOVA through each day since first dose at the daily geometric average prevalence-adjusted nAb $ID_{50}$ titre value after censoring participants for their second dose. †The daily Epanechnikov kernel-smoothed hazard functions were derived with an optimised window. From these the observed efficacy (dashed red line) was estimated as 100(1 – hazard ratio) (%), with two-sided bootstrapped 95% CIs (shaded red region) calculated from 1000 resamples with replacement. The number of at-risk participants over time and event times in SUPERNOVA are presented across the bottom for each arm. The comparator group includes participants who received tixagevimab–cilgavimab followed by placebo and those who received two doses of placebo. CI confidence interval, COVID-19 coronavirus disease 2019, $ID_{50}$ 50% inhibitory dilution, nAb neutralising antibody, N number of participants receiving each treatment, n number of participants with COVID-19 cases for each treatment, ToP threshold of protection.

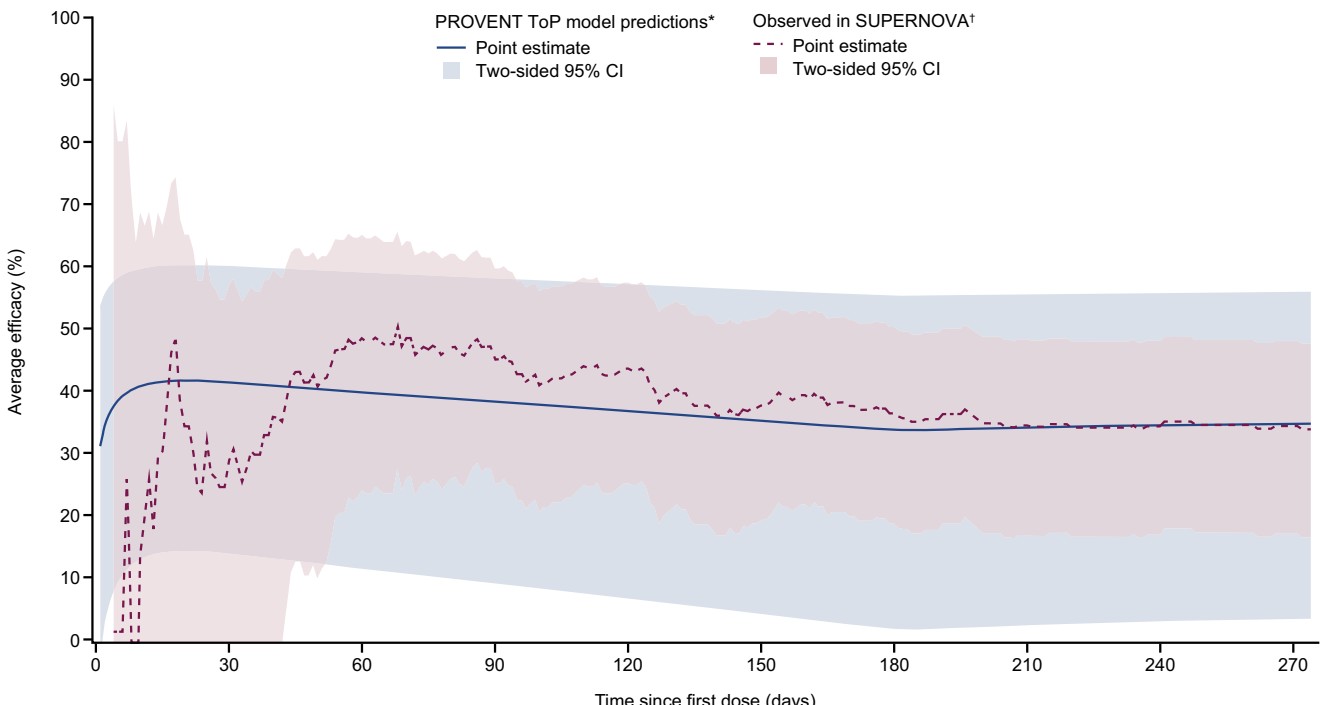

**Fig. 5 | External assessment of average overall efficacy from PROVENT ToP model using SUPERNOVA data.** *The PROVENT ToP model was developed against PROVENT data using a time-varying Cox model with adjustment for the treatment and its interaction with $\log_{10}$(prevalence-adjusted nAb $ID_{50}$ titres + 1). The ToP model is used to estimate average overall efficacy (solid blue line) and two-sided 95% CI (shaded blue region) through assessment at the average prevalence-adjusted nAb $ID_{50}$ titre value up to each day since first dose using SUPERNOVA study data. †Observed efficacy was defined based on actual intervention as 100(1 − relative risk) (%) of sipavibart versus comparator, where relative risk (dashed red line) and two-sided 95% CI (shaded red region) were evaluated with Poisson regression with robust variance, which includes treatment and randomisation stratification factors as covariates and adjusted for the follow-up time at each day since first dose. Randomisation stratification factors were COVID-19 vaccination status within 6 months before randomisation (Yes or No), SARS-CoV-2 infection within 6 months before randomisation (Yes or No), and tixagevimab–cilgavimab use within 12 months before randomisation (Yes or No). The comparator group includes participants who received tixagevimab–cilgavimab followed by placebo and those who received two doses of placebo. CI confidence interval, COVID-19 coronavirus disease 2019, $ID_{50}$ 50% inhibitory dilution, nAb neutralising antibody, SARS-CoV-2 severe acute respiratory syndrome coronavirus 2, ToP threshold of protection.

circulation and global monitoring of variants was a priority. It is fortuitous that this led to a wide range of predicted nAb $ID_{50}$ titres and corresponding efficacy estimates, allowing for adequate data to inform the efficacy-nAb titre relationship. As COVID-19 moves into an endemic phase, and countries scale back surveillance programmes, it is unlikely that future ToP studies will have access to the same quality of prevalence data. Under these methods, both overall and variant-defined efficacy predictions from the PROVENT ToP model align with those seen in the phase 3 SUPERNOVA study with a different mAb (Fig. 6); we therefore hypothesise that this would be generalisable to other settings.

Limitations include the following. Any error from estimating the parameters in the popPK model is not propagated through to the final ToP model. Prevalence data were not adjusted to account for delays between viral exposure and time of patient sample collection. However, event data in PROVENT and SUPERNOVA were reported as time of first symptoms, thus any misalignment will be negligible. Sipavibart in vitro $IC_{50}$ values for lineages with F456X were imputed to the upper limit of the assay. However, this method yielded conservative estimates of the observed F456L efficacies from SUPERNOVA (Fig. 6). This analysis makes several assumptions: that circulating variant prevalence is an adequate tool for quantifying exposure; and that variant-specific nAb ToP titres confer equivalent degrees of clinically relevant protection across variants. Future studies should aim to investigate the veracity of this variant-transportability assumption, as it underpins a ToP that can be applied to any SARS-CoV-2 variant. Additionally, surveillance data may contain sources of error, and countrywide data may

not reflect individual exposures impacting the categorisation of observed lineages into variants. However, it is expected that these sources of error will reduce precision but not introduce systematic bias.

The approach presented here allows efficacy to be assessed in the presence of multiple circulating variants that may change in prevalence over time, which better aligns with the current variant landscape. This is an advance over the method considered by Stadler et al.[13], which analysed efficacy against a single dominant variant. To assess this, the PROVENT ToP model was developed based on the dominant Delta variant, which had a similar in-vitro $IC_{50}$ to the Alpha variant that had dominated previously. This was done with and without an efficacy intercept, both resulting in poor estimation of efficacy (Supplementary Fig. S9).

There are several further applications of the work presented here. The ToP model could be used to expedite clinical evaluation of new mAbs by defining a target nAb level associated with a desired level of efficacy to be used as an immunobridging endpoint (e.g., show non-inferiority of predicted nAb $ID_{50}$ titres against a contemporary variant compared with a target nAb $ID_{50}$ titre associated with efficacy; Supplementary Methods). Furthermore, the ToP model could be used to estimate the duration of clinically relevant protection provided by an existing mAb against symptomatic COVID-19 due to a specific SARS-CoV-2 variant, using the predicted neutralisation titre curve (Supplementary Fig. S10). This is particularly beneficial given the relative ease of calculating the predicted neutralisation titre from popPK model predictions and in vitro $IC_{50}$ values. This may

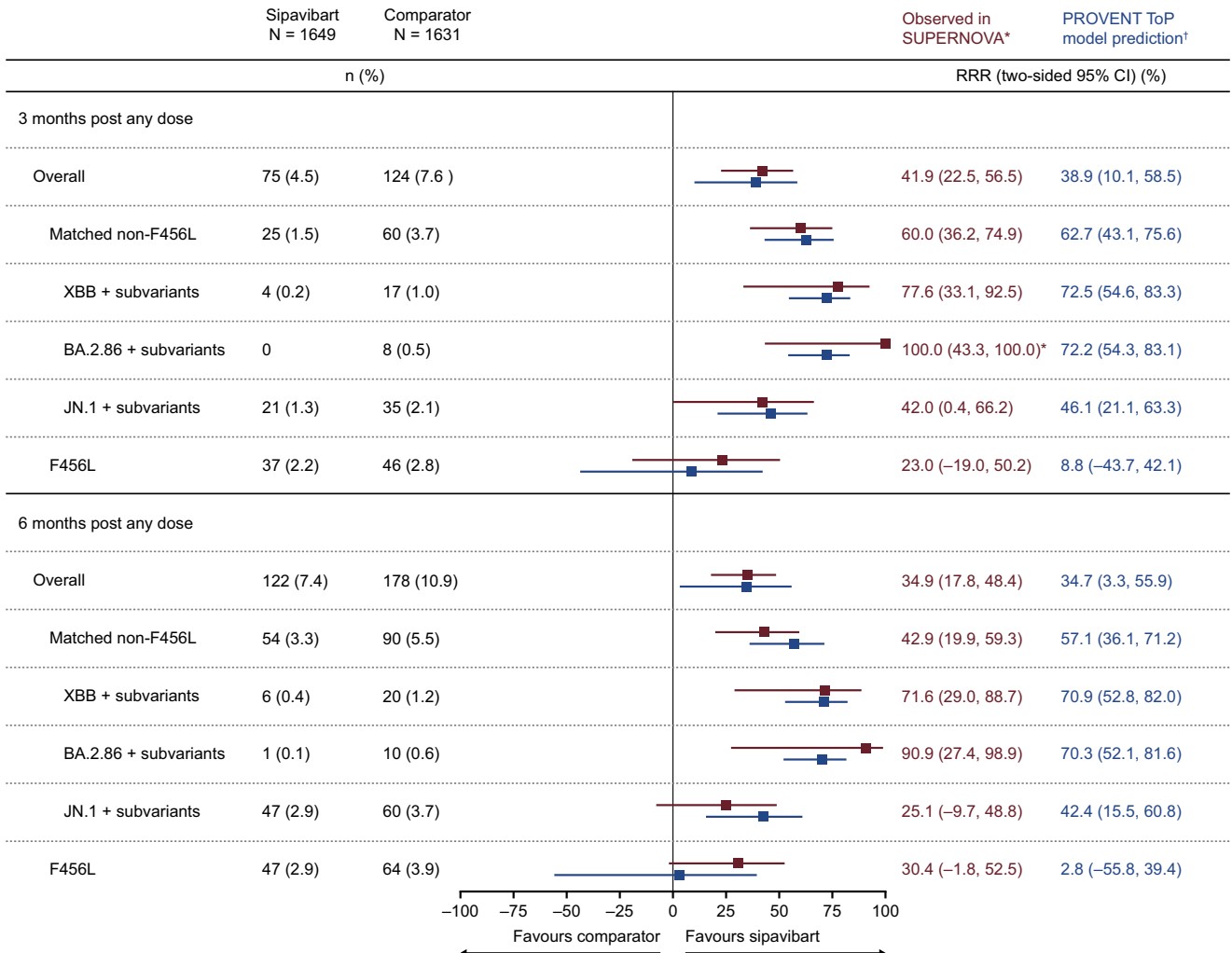

**Fig. 6 | Forest plot of SUPERNOVA observed efficacy versus PROVENT ToP model predicted efficacy based on SUPERNOVA data.** *Observed efficacy was defined based on planned intervention as 100(1 – relative risk) (%) of sipavibart versus comparator, where relative risk and two-sided 95% CI were evaluated with Poisson regression with robust variance, which includes treatment and randomisation stratification factors as covariates and adjusted follow-up time. Randomisation stratification factors were COVID-19 vaccination status within 6 months before randomisation (Yes or No), SARS-CoV-2 infection within 6 months before randomisation (Yes or No), and tixagevimab–cilgavimab use within 12 months before randomisation (Yes or No). †When events are not present in at least one arm, an exact approach is followed and 97.5% two-sided CI presented[10]. ‡The PROVENT ToP model estimates and covariance parameter estimates were used to predict efficacy (%) and two-sided 95% CI based on actual intervention through evaluating average nAb ID$_{50}$ titres over 90 days or 180 days post any dose according to the timepoint. For the overall endpoint, average prevalence-adjusted nAb ID$_{50}$ titres were considered after mapping GISAID prevalence data to at-risk participants for variants with ≥5% prevalence. For matched non-F456L and XBB+ subvariant

endpoints a similar approach was followed with exclusion of F456X variants (which are otherwise assumed an IC$_{50}$ of 1000), and inclusion of only XBB subvariants based on hedgehog variants that included a XBB Pango lineage, respectively. Daily serum mAb concentrations estimated from population pharmacokinetics were used to predict nAb ID$_{50}$ titres in at-risk participants based on IC$_{50}$ of 3.8, 83.1, and 1000 ng/mL for BA.2.86, JN.1 subvariant analyses and variants with F456L mutations, respectively. The comparator group includes participants who received tixagevimab–cilgavimab followed by placebo and those who received two doses of placebo. CI confidence interval, COVID-19 coronavirus disease 2019, IC$_{50}$ 50% inhibitory concentration, ID$_{50}$ 50% inhibitory dilution, GISAID Global Initiative on Sharing All Influenza Data, mAb monoclonal antibody, nAb neutralising antibody, N is the number of participants in the full analysis set without a positive RT-PCR test for SARS-CoV-2 at baseline by planned intervention, n is the number of participants with SARS-CoV-2 events for each endpoint by planned intervention, RT-PCR, reverse transcription-polymerase chain reaction, SARS-CoV-2 severe acute respiratory syndrome coronavirus 2, ToP threshold of protection.

allow for more confident and flexible evaluation of variant-targeted dosing regimens. The increase in tixagevimab–cilgavimab dose from 300 to 600 mg following the emergence of Omicron variants, against which tixagevimab–cilgavimab had reduced potency, was based on comparisons of popPK simulations to a theoretical target serum concentration for 80% viral neutralisation in the nasal lining fluid, a key expected site-of-action for prophylaxis[26]. This theoretical target was selected based on available viral dynamic modelling; increasing the dose from 300 to 600 mg was predicted to provide 4–6 months of protection against Omicron BA.4/5. However, this approach cannot predict the precise level of efficacy expected

with this change in dose. Using methodology presented here, we can define a ToP to tie a specific nAb titre to an efficacy estimate based on data, without any additional theoretical assumptions on the site of action of the mAb or the target level of viral neutralisation. This approach can therefore better inform rapid decision making (e.g., dose change in response to viral evolution) without needing to assess clinical efficacy or measure nAb titres from trial participants.

To conclude, an established ToP for a SARS-CoV-2–neutralising mAb could be used to extrapolate efficacy in pre-exposure prophylaxis of COVID-19 to new populations and/or new mAbs with comparable

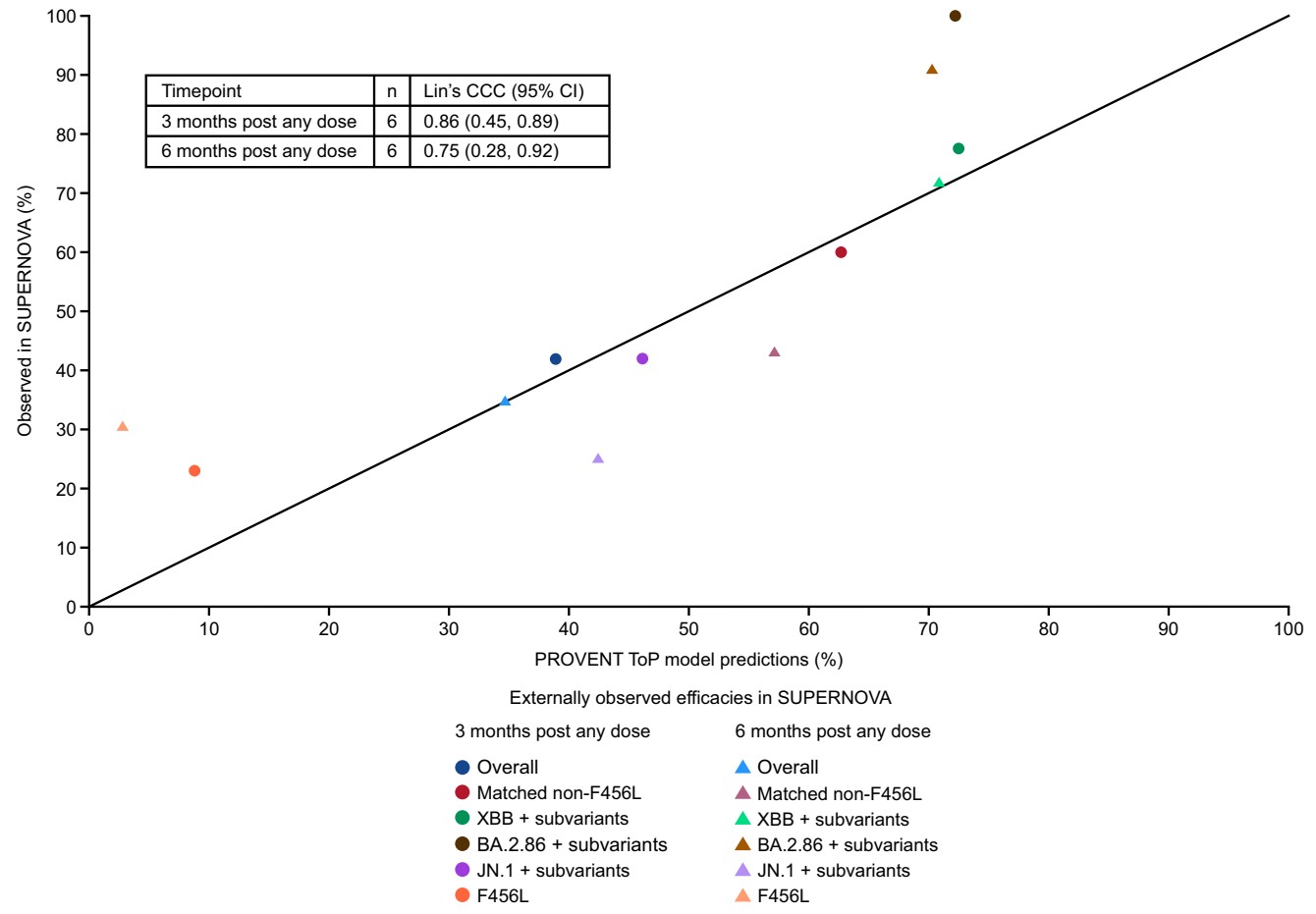

**Fig. 7 | Concordance between observed SUPERNOVA efficacies and PROVENT ToP model–derived efficacy predictions.** The observed SUPERNOVA efficacies and the respective PROVENT ToP model predictions from Fig. 6 are presented here and assessed for their concordance using Lin's CCC. CIs are bootstrapped from 1000 resamples. Reference line of $y = x$ plotted to visually assess the agreement. CCC concordance correlation coefficient, CI confidence interval, ToP threshold of protection.

mechanisms of action. Our findings add to the evidence that nAb titres are inversely correlated with symptomatic COVID-19 risk and directly correlated with efficacy, and contribute to the development and validation of a suitable methodological approach to estimating a ToP that corresponds to a given expected level of efficacy. This ToP can then be used as a benchmark for inferring clinically relevant protection against any specific SARS-CoV-2 variant or mix of variants.

## Methods
### Data sources
PROVENT (NCT04625725) was a randomised, double-blind, placebo-controlled, multicentre, phase 3 clinical trial that assessed the efficacy of a single intramuscular 300-mg dose of tixagevimab–cilgavimab compared with placebo (2:1 randomisation; $N = 5197$) for the prevention of symptomatic COVID-19[7]. The primary endpoint was a reduction in incidence of reverse transcription-polymerase chain reaction (RT-PCR)–confirmed symptomatic COVID-19 with tixagevimab–cilgavimab compared with placebo in SARS-CoV-2–naive participants who had not received a COVID-19 vaccine. PROVENT recruited adults (≥18 years of age) with an increased risk of inadequate COVID-19 vaccination response or SARS-CoV-2 exposure. Exclusion criteria included prior SARS-CoV-2 infection, receipt of a vaccine or biologic for prevention of COVID-19, or allergy to any component of tixagevimab–cilgavimab or placebo.

The SUPERNOVA main cohort study (NCT05648110) was a randomised, double-blind, multicentre, phase 3 clinical trial to assess the efficacy of sipavibart relative to comparator (1:1 randomisation; $N = 3335$) for the prevention of symptomatic COVID-19[10]. The study was ongoing at the time of this analysis. SUPERNOVA enrolled participants ≥12 years old who had a qualifying immunocompromising condition. Participants received a second dose of their original randomised study intervention (sipavibart or comparator) after 6 months. Originally the comparator arm comprised of only tixagevimab–cilgavimab; at the request of the regulatory authorities on June 14, 2023, the comparator was changed to placebo, leading to redosing in participants originally receiving tixagevimab–cilgavimab. The primary efficacy outcomes were symptomatic COVID-19 caused by any variant or by non-Phe456Leu–containing variants within 181 days of dosing. The primary analysis data cut was event driven, leading to longer follow-up for participants first receiving tixagevimab–cilgavimab (192.6 days) compared with those receiving placebo (146.2 days).

Both trials were conducted in accordance with guidelines from the Declaration of Helsinki, Council for International Organisations of Medical Sciences, and Good Clinical Practice Council for Harmonisation. The protocols were approved by the appropriate institutional review board or ethics committee at the study sites. The list of institutional review boards and ethics committees that approved the studies are available (Supplementary Files 1 and 2). All participants provided written informed consent. The institutional review boards/ethics committees were not required to approve the present modelling study. Since the purpose of the analysis is to better

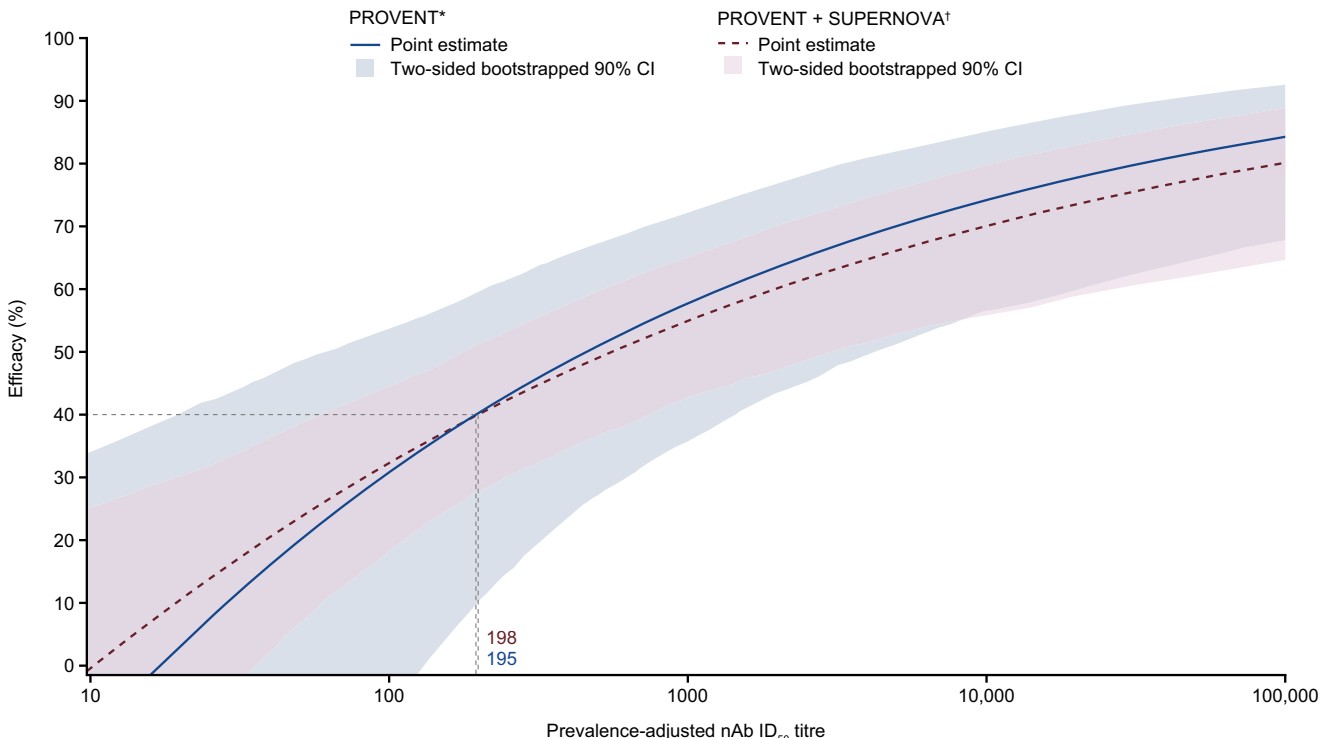

**Fig. 8 | ToP model sensitivity with inclusion of SUPERNOVA data.** *The PROVENT ToP model was defined as a time-varying Cox model adjusting for treatment and its interaction with $\log_{10}$(prevalence-adjusted nAb $ID_{50}$ titres + 1). †The PROVENT + SUPERNOVA ToP model was defined as a time-varying Cox model adjusting for treatment and its interaction with $\log_{10}$(prevalence-adjusted nAb $ID_{50}$ titres + 1) and stratifying the baseline hazard by study. Efficacy (PROVENT = solid blue line; PROVENT + SUPERNOVA = dashed red line) was calculated as 100(1 − hazard ratio) (%). The two-sided 90% CIs (PROVENT = blue shaded regions;

PROVENT + SUPERNOVA = red shaded region) were bootstrapped using 1000 resamples with replacement. The values where the point estimates met 40% were evaluated and presented in the droplines (short-dashed black lines). The 40% values are aligned with the go criteria from SUPERNOVA based on efficacy against symptomatic COVID-19 attributable to any variant. CI confidence interval, COVID-19 coronavirus disease 2019, GMT geometric mean titre, $ID_{50}$ 50% inhibitory dilution, nAb neutralising antibody, ToP threshold of protection.

characterise the efficacy of the mAb in the disease of interest, this is regarded as primary use of the data, and the participating patients directly consented to such use of the data in the informed consent form.

### Predicted daily serum mAb concentrations

Daily serum mAb concentrations were generated as input for this analysis based on the previously developed popPK models for tixagevimab–cilgavimab and sipavibart. Both models were two-compartment with linear elimination and first-order absorption following intramuscular administration. The tixagevimab–cilgavimab popPK model was fit to data from eight phase 1, 2, and 3 studies[26]. The sipavibart popPK model included all available data from ongoing sipavibart studies[10,27,28].

For PROVENT, daily serum mAb concentrations for each participant receiving tixagevimab–cilgavimab with at least one quantifiable post-baseline pharmacokinetic (PK) sample were generated using the tixagevimab–cilgavimab popPK model, with the tixagevimab–cilgavimab concentration calculated as the sum of concentrations of the individual mAbs[26]. Individual predictions (empirical Bayesian estimates) were generated from time of dosing through day 366 or receipt of a second dose of tixagevimab–cilgavimab, and were available for 3272 of the 3442 participants in the tixagevimab–cilgavimab arm, including all 63 participants with primary events.

In the main cohort of SUPERNOVA, PK samples were collected from approximately 600 participants in the sipavibart arm, with additional samples taken at all illness visits. Daily individual serum mAb concentration predictions were generated using the sipavibart popPK model. Of the 1649 participants who received sipavibart,

individual popPK predictions (empirical Bayes estimates, sampled daily) were generated for 768 participants with at least one available post-dose PK sample (at scheduled visits and/or illness visits). For the remaining 881 participants, popPK simulations were generated based on the individual participant dosing records and baseline characteristics[10]. The output was the mean of 10 replicates per individual, conducted by sampling from between-subject variability. Of 1649 participants in the sipavibart arm, 122 had a reported SARS-CoV-2 case per the primary endpoint definition for overall efficacy, 115 of whom had individual popPK predictions generated, and seven of whom had no available post-dose PK samples and therefore had PK profiles estimated based on simulation.

### In vitro potency and serum nAb $ID_{50}$ titre assessments

In vitro $IC_{50}$ of tixagevimab–cilgavimab and sipavibart were assessed, along with observed serum nAb $ID_{50}$ titres from sera collected from participants in PROVENT in the same pseudovirus neutralisation assay (PhenoSense SARS-CoV-2 nAb assay; Monogram Biosciences, San Francisco, CA). The PhenoSense SARS-CoV-2 nAb assay is based on previously described methodologies[29,30] using a lentivirus system, where human immunodeficiency virus (HIV) pseudovirus virions expressed the SARS-CoV-2 spike protein from the variant of interest. HEK293 cells were transfected with a HIV genomic vector that contained a luciferase report gene plus an envelope expression vector carrying the SARS-CoV-2 spike protein open reading frame. Neutralising titres or $IC_{50}$ values were measured by assessing inhibition of luciferase activity, following preincubation of pseudovirions with serial dilutions of sera from trial participants or from in vitro dilutions of mAbs.

**Table 2 | SARS-CoV-2 variants considered in threshold analysis**

| Study | WHO SARS-CoV-2 name | Pango lineage (Hedgehog)[a] | In vitro IC$_{50}$ (ng/mL) |
|---|---|---|---|
| PROVENT | Alpha | B.1.1.7 | 2.1 |
| | Beta | B.1.351 | 5.6 |
| | Wuhan | – | 2.2 |
| | BA.1 | B.1.1.529.1 | 171.1 |
| | BA.2 | B.1.1.529.2 | 9.8 |
| | Delta | B.1.617.2 | 2.2 |
| | BA.1.1 | B.1.1.529.1.1 | 466.0 |
| | Eta | B.1.525 | 9.5 |
| | Gamma | P.1 | 1.3 |
| | Kappa | B.1.617.1 | 1.9 |
| | Lambda | C.37 | 1.2 |
| | Mu | B.1.621 | 17.2 |
| | BA.4/5 | B.1.1.529.4 | 69.4 |
| | Iota | B.1.526 | 3.3 |
| | Zeta | P.2 | 5.4 |
| | Epsilon | B.1.427/B.1.429 | 1.3 |
| SUPERNOVA | Omicron | BA.4 (B.1.1.529_1) | 4.7 |
| | | BA.2 (B.1.1.529_2) | 10.7 |
| | | XBB.1 (B.1.1.529_3)[b] | 3.6 |
| | | FL.1.5.1 (B.1.1.529_6)[c] | 1000 |
| | | BA.1.1 (BA.1.1_1) | 4.6 |
| | | BA.1 (BA.1_1) | 5.4 |
| | | BA.2.12.1 (BA.2.12.1_1) | 7.9 |
| | | JF.1 (BA.2_21)[c] | 1000 |
| | | HV.1 (BA.2_22)[c] | 1000 |
| | | HK.3 (BA.2_3)[c] | 1000 |
| | | BA.2.75 (BA.2.75) | 25 |
| | | BA.2.86 (BA.2.86_1) | 3.8 |
| | | XBB.1.5 (BA.2_1)[b] | 5.8 |
| | | EG.5.1 (BA.2_2, BA.2_8)[c] | 1000 |
| | | EG.5 (BA.2_49, FE.1.1.3, FL.19.1)[c] | 1000 |
| | | JN.1 (BA.2_4) | 83.1 |
| | | XBB.2.3.2 (BA.2_44)[b] | 5.6 |
| | | XBB (BA.2_5)[b] | 3.8 |
| | | GW.5 (BA.2_6)[c] | 1000 |
| | | GK.1 (BA.2_7)[c] | 1000 |
| | | JD.1.1 (BA.2_10)[c] | 1000 |
| | | BA.2.75.2 (BA.2_9) | 9.7 |
| | | BA.4.6 (BA.4.6_1) | 14.5 |
| | | BA.5.9 (BA.5.9) | 4.7 |
| | | BF.7 (BA.5_1) | 3.8 |
| | | BQ.1 (BA.5_2) | 11.6 |
| | | BN.1 (BN.1_1, BN.1.2.5, BN.1.3.12) | 8.3 |
| | | BN.1.3 (BN.1.3.3) | 7.9 |
| | | BQ.1.3 (BQ.1.3_1) | 13.4 |
| | | BQ.1.1 (BQ.1_1) | 9.2 |
| | | BR.2 (BR.2_1) | 21.2 |
| | | DV.7.1 (DV.7.1_1, DV.7.1.2)[c] | 1000 |
| | | XBB.16.1 (XBB.1.16.1_1)[b] | 6.3 |
| | | XBB.1.16.24 (XBB.1.16.24)[b] | 3 |

**Table 2 (continued) | SARS-CoV-2 variants considered in threshold analysis**

| Study | WHO SARS-CoV-2 name | Pango lineage (Hedgehog)[a] | In vitro IC$_{50}$ (ng/mL) |
|---|---|---|---|
| | | XBB.1.5.1 (XBB.1.5.1)[b] | 4.6 |
| | | XBB.1.5.10 (XBB_2)[b,c] | 1000 |
| | | XBB.1.19.1 (XBB.1_2)[b] | 4.5 |
| | | FD.2 (XBB.1_3)[b] | 4.5 |
| | | XBB.2.3.2 (XBB.2.3.2_1)[b] | 5.6 |
| | | XBB.2.3 (XBB.2.3_1)[b] | 3.4 |
| | | XBB.1.16 (XBB_1)[b] | 1.3 |
| | | Other with F456X mutation | 1000[d] |
| | | Other with L455X mutation | 83.1[d] |
| | | Other without L455X/F456X mutations | 7.1[d] |

The 16 variants considered for the PROVENT threshold analysis were those with ≥1% prevalence over follow-up and where in vitro IC$_{50}$ measurements were available (Monogram Biosciences). The variants considered in the SUPERNOVA threshold analysis are those with ≥5% prevalence over follow-up.

IC$_{50}$ 50% inhibitory concentration, nAb neutralising antibody, SARS-CoV-2 severe acute respiratory syndrome coronavirus 2, WHO World Health Organisation.

[a]Hedgehog lineages are displayed for SUPERNOVA in brackets with one representative Pango lineage with available IC$_{50}$ shown for each hedgehog lineage. Due to Hedgehog utilising nucleotide sequences, some amino acid spike sequences tested for IC$_{50}$ may map to multiple hedgehog lineages.

[b]This family of variants is considered in subsequent XBB + subvariant analyses.

[c]These variants all contained an F456L mutation with IC$_{50}$ measured above the upper assay limit of 1000 ng/mL.

[d]Where IC$_{50}$ was not measured for a variant, variants with F456X and L455X mutations are imputed to a value of 1000 ng/mL (equal to the upper assay limit) and 83.1 ng/mL (consistent with JN.1), respectively, otherwise imputed to 7.1 ng/mL, which represents an average of known in vitro IC$_{50}$ values for sipavibart against variants that do not have either F456X or L455X mutations.

Following from previous work[21], the predictive accuracy of nAb ID$_{50}$ titres derived from serum concentration and in vitro IC$_{50}$ values for PROVENT was explored using a small subset of the observed serum nAb ID$_{50}$ titres measured in the same assay as the in vitro IC$_{50}$ values (Supplementary Methods).

### Variant prevalence data

Full-length SARS-CoV-2 genomic sequencing data from the Global Initiative on Sharing All Influenza Data (GISAID) database were used to infer country-specific variant prevalence[31,32]. For PROVENT, variants, except for Omicron, were grouped by their World Health Organisation (WHO) nomenclature and regional daily prevalence was calculated per WHO group (Table 2). Omicron variants were grouped by major subvariants BA.1, BA.1.1, BA.2, BA.3, and BA.4/5 (with BA.4 and BA.5 sharing identical spike sequences). Variants with ≥1% daily prevalence were reported and those without a WHO label were omitted (4.7% of total sequenced data). During SUPERNOVA, SARS-CoV-2 surveillance sequencing was significantly reduced, leading to higher variability in prevalence percentages, especially for less prevalent lineages; variants were therefore limited to those with daily prevalence >5%. Additionally, due to fewer lineage IC$_{50}$ values available, Pango lineage assignment based on spike sequence only (Hedgehog) was utilised[33]. This approach allowed IC$_{50}$ values to be assigned to all lineages sharing an identical spike sequence. Measured IC$_{50}$ values against the virus with a matched spike sequence of a given lineage were used where possible. Lineages without available measured IC$_{50}$ values were categorised into three groups for IC$_{50}$ value assignment: with mutation at 455 (L455X); with mutation at 456 (F456X); without mutation at 455 or 456. Lineages with L455X were assigned an IC$_{50}$ value of 83.1 ng/mL,

corresponding to JN.1-containing L455S. Lineages with F456X were assigned an $IC_{50}$ of 1000 ng/mL, corresponding to the upper limit of the assay, consistent with loss of potency to variants containing F456L based on measured $IC_{50}$ values against multiple variants containing this mutation. Lineages without either of these mutations were assigned a surrogate $IC_{50}$ value of 7.1 calculated based on the average $IC_{50}$ values of variants without either L455X or F456X. On-study prevalence is presented in the Supplementary Methods.

### Prevalence-adjusted in vitro potency

Variant prevalence data were combined with individual-level data by region (country for PROVENT/continent for SUPERNOVA) and calendar date[31,32]. Once mapped, the prevalence-adjusted $IC_{50}$ was derived as prevalence-weighted geometric mean of $IC_{50}$ values. For each participant and day, we calculated a prevalence-adjusted $IC_{50}$ as a weighted geometric mean of variant-specific $IC_{50}$ values (Table 2), based on regional variant prevalence, reflecting each participant's exposure to variants circulating at that time. Additional details are provided in the Supplementary Methods.

### Prevalence-adjusted predicted daily nAb $ID_{50}$ titres

Daily prevalence-adjusted predicted nAb $ID_{50}$ titres were generated from the predicted daily mAb (tixagevimab–cilgavimab or sipavibart) serum concentrations and prevalence-adjusted in vitro $IC_{50}$ values: serum mAb concentration (ng/mL)/prevalence-adjusted $IC_{50}$ (ng/mL). The predicted nAb $ID_{50}$ titre formula has been shown to align well with observed nAb $ID_{50}$ titres following tixagevimab–cilgavimab administration across different SARS-CoV-2 variants[26]. This method of deriving predicted nAb $ID_{50}$ titres provides an estimate that is specific to the administered mAb, independent of baseline nAb titres related to previous exposure or vaccination status or changes in nAb titres related to vaccination or SARS-CoV-2 infection during the clinical trial (consistent with the while-on-treatment primary estimands). Further, through prevalence-adjustment it is hypothesised that the associated nAb titre-efficacy relationship is independent of the SARS-CoV-2 variant.

### A ToP for clinically relevant efficacy derived from the relationship with prevalence-adjusted nAb $ID_{50}$ titres

This model is referred to as the PROVENT ToP model. A Cox model with time-varying covariates was fit to RT-PCR–confirmed symptomatic COVID-19 through day 366, with covariates for treatment and its interaction with an intercept term and $\log_{10}$(prevalence-adjusted nAb $ID_{50}$ titres + 1). This model is similar to that used to establish the associated efficacy–nAb titre relationship for HIV[34]. Time was measured from randomisation and participants were censored for unblinding for consideration of, or for receipt of, a COVID-19 vaccine[7].

Four time-varying Cox models were considered. The first was a single-parameter time-varying Cox model that is supported by the biologically plausible assumption that the nAb-efficacy curve passes through the origin (no intercept). The second was a two-parameter model that relaxes this assumption to allow greater model flexibility (efficacy intercept). The third also relaxes this assumption whilst adjusting for baseline covariates (efficacy intercept + baseline covariates). The fourth increased from zero efficacy once a threshold level of nAb $ID_{50}$ titres was achieved (nAb $ID_{50}$ titre intercept). The second model was chosen. Further details of the selected ToP model are presented in the Supplementary Methods. Once the appropriate parameterisation was selected further evaluations of the appropriate transformation of prevalence-adjusted nAb $ID_{50}$ titres were evaluated and the transformation $\log_{10}$(prevalence-adjusted nAb $ID_{50}$ titres + 1) was applied (Supplementary Methods).

### Validation of the PROVENT ToP model

Data from SUPERNOVA was used to externally validate predictions from the PROVENT ToP model using the three approaches described.

As it may be of interest to examine how mAb efficacy changes over time following administration (i.e., to assess relationship with falling serum mAb concentration), an assessment of whether the PROVENT ToP model could predict overall (i.e., attributable to any SARS-CoV-2 variant) instantaneous efficacy from SUPERNOVA was conducted. The PROVENT ToP model was evaluated at the daily geometric mean of the prevalence-adjusted nAb $ID_{50}$ titres to generate daily instantaneous efficacy predictions and two-sided 95% CIs for the SUPERNOVA study. To compare this with observed data, daily Epanechnikov kernel-smoothed hazard functions were derived with an optimised window, and the observed efficacy was estimated as 100(1 – hazard ratio) (%). Further details are provided in the Supplementary Methods.

In clinical trials, efficacy is commonly aggregated through to data cutoff. Therefore, an assessment of whether the PROVENT ToP model could predict the overall average efficacy from SUPERNOVA was conducted cumulatively for each day since first dose. The geometric mean of prevalence-adjusted nAb $ID_{50}$ titres up to each day since first dose from SUPERNOVA were evaluated by the PROVENT ToP model. These were visually compared with efficacy that would have been reported at each day since first dose under the pre-defined efficacy analysis from SUPERNOVA[10], referred to here as observed efficacy. Further details are provided in the Supplementary Methods.

Given the prevalence-standardised approach used in the PROVENT ToP model, it is of interest to assess the ability of the model to predict efficacy against a specific variant or a mixture of variants. Planned analyses from SUPERNOVA included the average efficacy of overall, matched non-F456X mutation variants, subvariant-specific estimates, and F456X mutation variants at 3-month and 6-month timepoints relative to any dose[10]. The geometric mean of the variant-specific or prevalence-adjusted nAb $ID_{50}$ titres over the two timepoints were evaluated by the PROVENT ToP model to provide predictions corresponding to these efficacy estimates. External predictive accuracy was assessed using Lin's CCC separately for the two timepoints. Further details are provided in the Supplementary Methods.

### Model robustness to SUPERNOVA data

The model was updated with the inclusion of both PROVENT and SUPERNOVA data to assess the PROVENT ToP model robustness. This model is henceforth referred to as the PROVENT + SUPERNOVA ToP model. The time-varying Cox model was fit to the pooled data with a study-specific stratification factor to allow the baseline hazard to vary. Further details are provided in the Supplementary Methods.

### Conversion to international units (IU/mL)

The same pseudovirus neutralisation assay (PhenoSense SARS-CoV-2 nAb assay; Monogram Biosciences, San Francisco, CA) was used for measuring mAb $IC_{50}$ values and serum nAb $ID_{50}$ titres. Mean nAb $ID_{50}$ titres generated by the replicate testing of the WHO first international standard, National Institute for Biological Standards and Control 20/136 using the PhenoSense SARS-CoV-2 nAb Assay (Wuhan D614 variant, 24 measurements across multiple vials, operators, and days) were used to calculate a conversion factor of 0.1428 that enabled the conversion of nAb median infectious dose titre to IU/mL. Although this standard was established for the Wuhan variant at the start of the pandemic, neutralisation titres across variants are frequently presented on the same titre scale and the titre shift (fold-change from one variant to another) is considered reflective of change in neutralising ability of sera or mAbs. Similarly in this work, we combined titres across multiple variants based on variant prevalence. The format of pseudovirus neutralisation assay used here was the same across multiple variants, except for the change in SARS-CoV-2 spike sequence, so any change in

titre was the result of change in neutralising ability of a mAb or sera. The same factor could therefore be applied to convert the resulting titre threshold to IU/mL.

## Reporting summary

Further information on research design is available in the Nature Portfolio Reporting Summary linked to this article.

## Data availability

The data underlying the findings described in this manuscript consist of anonymised patient-level datasets from clinical trials, which cannot be freely shared to protect patient privacy in accordance with General Data Protection Regulation (GDPR) and other applicable local legislation, and subject to the content of informed consent forms, access can be obtained in accordance with AstraZeneca's data sharing policy described at https://www.astrazenecaclinicaltrials.com/our-transparency-commitments/. Data for studies directly listed on Vivli can be requested through Vivli at www.vivli.org. Data for studies not listed on Vivli could be requested through Vivli at https://vivli.org/members/enquiries-about-studies-not-listed-on-the-vivli-platform/. AstraZeneca Vivli member page is also available outlining further details: https://vivli.org/ourmember/astrazeneca/. Data will be made available upon approval of the request and signature of the Data Usage Agreement. Typically, data will be available up to 1 year starting on the date access was granted.

## Code availability

All analyses were conducted using the commercially available SAS version 9.4 or higher (SAS Institute, Cary, NC, USA). Custom code used for fitting the ToP model described in this manuscript as well as a sample dataset for running the code may be obtained from the Github repository at https://github.com/AstraZeneca/ToPModel/[35].

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

## Acknowledgements

We gratefully acknowledge all the authors from the originating laboratories responsible for obtaining the specimens and the submitting laboratories where genetic sequence data were generated and shared via GISAID, on which this research is based. In vitro $IC_{50}$ and observed serum nAb $ID_{50}$ titres were measured at Monogram Biosciences (San Francisco, CA, USA). The authors thank Jonathan Toma, Paul Theobald, Carmeliza Santos, Terri Wrin, Chris Petropoulos and others at Monogram Biosciences (San Francisco, CA, USA) for continuous support of next-generation sequencing and neutralisation assay analyses. For sipavibart, the majority of $IC_{50}$ values were measured in pseudovirus assays conducted by Monogram Biosciences as part of the HHS/USG COVID Therapeutics Response efforts. Tixagevimab, cilgavimab, and sipavibart serum concentrations were measured at PPD (Richmond, VA, USA). The authors also acknowledge Rebecca A. Bachmann, PhD, for valued contributions to the manuscript development. Medical writing support was provided by Lorna Forse, PhD, and editorial support was provided by Jess Fawcett, BSc, of Core (a division Prime, London, UK), supported by AstraZeneca according to Good Publication Practice guidelines. AstraZeneca was involved in the study design, data collection and analysis, and review of manuscript drafts. Funding: this analysis was funded by AstraZeneca and includes data from the PROVENT trial that was funded by AstraZeneca and the United States Government.

## Author contributions

R.E., S.M., S.S., A.A.A., L.C., B.A., K.S. and T.S.C. contributed to the conceptualisation and design of the manuscript. R.E., S.M., B.A., S.S., and J.P. were responsible for the data analysis and validation. All authors (R.E., S.M., B.A., A.A.A., L.C., J.L.P., M.T.E., L-J.C., I.H., T.V., J.P., O.S., K.S., T.W., T.S.C., D.F., P.B.G. and S.S.) interpreted and critically reviewed drafts and provided approval to submit.

## Competing interests

R.E., B.A., A.A.A., L.C., M.T.E., L-J.C., I.H., T.V., J.L.P., J.P., O.S., K.S., T.W., T.S.C. and S.S. are employees of, and may hold stock in, AstraZeneca. S.M. is an employee of MMS Holdings Inc. P.B.G. is a member of the AstraZeneca Advisory Board through a contract to his institution. D.F. reports no conflicts.
