## [Peer Review file · Nature Communications]

A SARS-CoV-2 variant-adjusted threshold of protection model for monoclonal antibody pre-exposure prophylaxis against COVID-19

Corresponding Author: Dr Seth Seegobin

Version 0:

Reviewer comments:

Reviewer #1

(Remarks to the Author)

Edge et al. use data from the PROVENT trial to predict the neutralizing antibody (Ab) titer from the serum concentration of the monoclonal antibody (mAb) Evusheld and predict the protective effectiveness of the mAb and the duration of protection against different SARS-CoV-2 variants. The main results are a prevalence-adjusted and fold-IC50 titer of 458 corresponding to at least 50% protective efficacy (~65 IU/ml) and predictions of the duration of protection above 50% against different variants (and in the presence of multiple variants by prevalence-adjusted titers).

The main strength of this work is the use of individual-level serum concentration data. The manuscript is well written and clear and the methodology is appropriate (besides concerns about the suitability of the model, see comment 4). However, the results seem to be lacking sufficient relevance and novelty for a journal like Nature Communications, a more field-specific journal may be more appropriate. Moreover, the results (model predictions) are not validated.

Major comments:

1. Lacking relevance:

Monoclonal antibodies are largely no longer used to treat COVID-19. Instead, antivirals have been shown to be much more robust to the emergence of new variants and are highly effective. Here, the authors present an analysis of one trial of one particular mAb (that is no longer used to treat COVID-19), against variants that are no longer circulating. Also, although COVID-19 treatments are primarily for the immunocompromised, their data is not clearly related to such populations (as the PROVENT trial also includes participants with increased risk of exposure, l. 121). It is unclear that there is a use case for the current model proposed by the authors.

2. Lacking novelty:

Serum concentrations of mAbs have previously been related to neutralizing titers [1]. Inclusion of different variants in the same model by scaling nAb titers has been reported previously [2,3] and the 'variant transportability assumption' (l. 349) of correlates of protection has been investigated for early variants and SARS-CoV-2 vaccination [2]. The authors have previously published a quantitative relationship between mAb concentration and protection [4]. A relationship between protective effectiveness of mAbs and the nAb titer has also been shown previously and used to predict the duration of protection, with the inclusion of variations the neutralizing potency (IC50) against different variants [3]. As the authors state, pemivibart received emergency use authorization based on an immunobridging strategy (l. 360-363), so how does their work add to existing and already applied immunobridging strategies for mAbs?

How does this work compare to existing approaches, what is the novelty, and what is the added benefit compared to existing methods? The research context is not sufficiently considered and discussed in the manuscript. In this reviewer's opinion, after such a discussion of the existing literature it is difficult to see particular novelty in the current study.

3. Lack of validation of results:

The authors make a problematic statement as to the validation of their model prediction by stating that their results are "in agreement with observed data from the PROVENT final analysis" (l. 290-291) which is not surprising as this was the data used to calibrate the model. Since there are already existing works doing a similar analysis to that reported here (see above), it seems external validation might be possible, but the authors do not compare their model with existing models relating protective effectiveness of mAbs to Ab titers (e.g., [4] and [3]). The authors openly admit the need for validation in the discussion, but this does not in any way reduce the need to validate results.

4. Model choice:

The authors use a model that predicts negative effectiveness for low nAb titers which is biologically implausible (l. 299). They compare this model with another model that predicts high efficacy at low titers that "may not be biologically feasible" (Supplement, l. 146). Model predictions of the threshold of protection differ greatly (458 vs 160) which reduces the confidence in either estimate. Why are the authors not considering another model that is biologically plausible with non-negative effectiveness and allowing for a steeper increase in effectiveness to fit low effectiveness at low titers and higher effectiveness at higher titers? Generally, the conclusions of a study should not be so dependent on modelling assumptions that are clearly implausible.

References

1. Clegg LE, et al. Serum AZD7442 (tixagevimab–cilgavimab) concentrations and in vitro IC50 values predict SARS-CoV-2 neutralising antibody titres. *Clinical & Translational Immunology* 13, (2024).
2. Cromer D, et al. Predicting vaccine effectiveness against severe COVID-19 over time and against variants: a meta-analysis. *Nat Commun* 14, 1633 (2023).
3. Stadler E, et al. Monoclonal antibody levels and protection from COVID-19. *Nat Commun* 14, 4545 (2023).
4. Follmann D, et al. Examining protective effects of SARS-CoV-2 neutralizing antibodies after vaccination or monoclonal antibody administration. *Nat Commun* 14, 3605 (2023).

Reviewer #2

(Remarks to the Author)

Positive feedback. The text is short and to the point. Easy to read. Rationale and most details are well explained and straightforward to follow.

Major comments

The authors make several assumptions to arrive to the conclusion that "a nAb titre of 458 (~65 IU/ml) corresponded to $\geq 50\%$ efficacy". Is this conclusion robust? Per my reading of the paper, I do not see analyses that illustrate robustness (and translation potential to other settings/studies). Critical assumptions seem to be the following: 1) nAb decay is the same in all individuals -> but obviously it is not -> if such variability is introduced how is the number 458 changed? 2) authors use log10 Ab as predictor. This seems weird, it should be Ab titer as mechanistically Abs and not log of Abs should be predictors of protection. 3) Authors normalize Ab titer by IC50 from a specific assay. Why IC50? Would the model without normalization fail to fit the data? What about IC10? IC90? IC99? 4) Ab titers were weighted using log titers. Why not linear?

Data shown in Suppl Fig 3 (probably should be in main text as these are original data) are interesting but it is hard to understand why treated individuals accumulate infections at much faster rate after day 156. This may indicate biases in the data. Also, the rate of accumulation of cases in treated cases does not speed up but is expected due to individuals losing protection at different rates as Abs decay differently in different individuals. This contradiction must be explored.

The key element of many studies of Abs and protection against COVID19 include some form of normalization of Ab titers, including those against the variants (e.g., 37507368). The idea seems reasonable but I have not seen much analysis that shows that such normalization is important, i.e., alternative models fail to fit the data. Another big concern is how to translate predicted numbers (e.g., 458) to other studies who do not use proprietary systems to measure nAb titers. This must be addressed.

I find it unacceptable that the data from the trials are not freely available but require requests to the company. The company should be able to anonymize the data and make them available as a requirement for publication.

Methodology of the modeling must be moved to main text with ALL equations numbered.

The model with intercept is useful but that it gives negative efficacy raises concern. Why not use a model that does not allow for negative values but does have intercept. That would be a deviation from standard hazard models but authors could devise a method based on likelihood to use a user-defined model to fit the model to data.

Minor comments

Figures need improvements. For multi-panel figures, each panel should be labeled as A/B/C (or similar), e.g., Figure 1.

It is impossible to see different line styles in Figure 1. Perhaps make a multipanel figure? Or use log-scale in A? Fig 1B is also impossible to follow -> perhaps making figure larger in y axis could help.

I do not understand Figure 2 -> why are there only 3 points for titers? Is the model fit here good? The fit does not seem good to me, averages are not explained well.

Are Figs 3&4 that useful for main text? Fig 4 is assumption, and Fig 3 is hard to interpret. Can you provide statistics to show that two models provide similar predictions besides a visual one?

Fig S1 -> what are the relationships here? Authors should fit the hill-based relationship and report hill coefficient -?> it seems that for some variants, relationship is not linear (e.g., BA.2) -> Why?

The extended Cox model (page 7 in supplement) was used -> is this a good model? Which alternative models did you try and why other models failed?

Fig S3 -> I do not understand log(t) value of 4. Please use log-scale but numerics that could be understood by humans.

I did not see a model for nAb dynamics. It should be listed along with estimated parameters. Did authors use nonlinear mixed effects to predict nAb kinetics?

Reviewer #3

(Remarks to the Author)

Edge et al. describe the development of a model to determine a neutralizing antibody (nAb) threshold of protection for monoclonal antibodies targeting SARS-CoV-2 in the context of pre-exposure prophylaxis against COVID-19. Using data from the PROVENT clinical trial of the mAb combination AZD7442, the authors determined that a nAb titer of 458 (~65 IU/mL) corresponded to >50% efficacy against symptomatic COVID-19. This model, based on predicted nAb titers and variant prevalence, was validated using data from multiple SARS-CoV-2 variants. The authors' conclusions are well-supported and offer valuable insights into the neutralizing antibody levels required for protection against COVID-19, potentially aiding regulatory approval of SARS-CoV-2 mAbs based on surrogate endpoints and reducing the need for large efficacy trials for new variants. However, the manuscript would benefit from additional data or discussion, as outlined in my comments below.

1. Line 34. "This study identified a threshold value for neutralising antibody titres (nAb) associated with clinically relevant protection against symptomatic COVID-19 for vulnerable populations". This statement is a bit misleading, as it suggests that the trial population was comprised of immunocompromised individuals which was not exclusively the case.
2. Were there enough cases in the clinical study to determine a threshold nAb titer for protection against severe disease or asymptomatic infections?
3. Did the authors measure mucosal antibody in nasal or oral swabs to understand levels of mucosal Abs required for protection against symptomatic disease? If not, the authors should comment that levels of nAb in the upper respiratory tract are likely responsible for the observed protection which will only be a small fraction of the serum Ab level given low efficiency of serum Ab transudation. This is probably the reason why relatively high concentrations of serum nAb are required to protect against upper respiratory disease.
4. The study population was SARS-CoV-2 naïve, which differs from the immune status of the majority of the global population today. Can the authors comment on how the nAb COP curve might shift in the setting of pre-existing immunity? For example, is it possible that T cell immunity might lower the required nAb threshold associated with 50% protection?
5. It would be helpful to also indicate nAb titers in IU/ml in figure 2 and to indicate the titer require for 70 or 80% protection which is the more likely target for future monoclonal therapies.
6. The raw data used for the COP modeling should be provided as a supplementary excel sheet.

Version 1:

Reviewer comments:

Reviewer #1

(Remarks to the Author)

In their revised manuscript Edge et al. include additional data from the SUPERNOVA study to validate their model. The main strengths of this manuscript are the use of individual-level serum concentration and infection data from two separate studies to establish and validate a correlate of protection for monoclonal antibody prophylaxis.

My previous comments concerned the lack of relevance, novelty, validation of results, and the model choice. In their updated manuscript, Edge et al. have added validation of their results by adding data from a second study and better explained the relevance and research context and highlighted that the main novelty lies in the use of individual data and the use of the prevalence-weighted mean IC50s to account for multiple variants circulating simultaneously.

The authors have sufficiently addressed my previous comments and I have only some minor comments on the updated version of the manuscript.

Minor comments:

1. L. 223: for the SUPERNOVA study, some IC50s were imputed at the upper assay limit of 1,000 ng/mL. Does this imputation affect the model estimates, i.e., if they were imputed at, e.g., 1,500 or 2,000 ng/mL, would that affect the conclusions and the comparison with the PROVENT ToP model?
2. L. 246: The authors state that their method is "independent of baseline nAb titers" but provide no evidence or references for this statement. In the PROVENT study the control group's titers were imputed to be zero and in the SUPERNOVA data comparator titers appear to also be negligibly low in the comparator group. However, e.g. the RECOVERY study showed different effectiveness of mAbs in seropositive and seronegative groups [1] and not all immunosuppressed patients have negligibly low titers. Would the method hold for an immunosuppressed cohort with low but non-zero titers?
3. The sentence in lines 482 to 484 was unclear to me.
4. Fig. 7: the caption mentions CIs, but there were no CIs in the figure.

References

1. RECOVERY Collaborative Group. Casirivimab and imdevimab in patients admitted to hospital with COVID-19 (RECOVERY): a randomised, controlled, open-label, platform trial. *The Lancet* 399, 665-676 (2022).

Reviewer #2

(Remarks to the Author)

Major comments

The two models of how nAb titers relate to protection are apparently inadequate. Simpler model (no intercept) does not predict protection adequately (over-estimates) and two parameter model predicts negative protection at low nAb titers. Claiming that a better fit model is correct is false as it is biologically incorrect. You need to come up with the 3rd model that would be biologically plausible and predicts accurately protection levels. For example, a model with no intercept but with $n \neq 1$. Alternative you could modify your 2 parameter model to predict 0 protection until $nAbs \leq A_{min}$, and increase after that. Unless you have a model that makes sense and fits the data well, the work is unfinished.

You used methodology to predict nAb levels using combination of IC50 and variant prevalence (E2). This is somewhat similar to what has been used by the Davenport group. But how do we know this expression works really better than anything else? You did not show any alternatives that do not explain the protection data. Again, per rigorous science is not enough to show what seem to work, it is critical to show what does not work. What is you don't use log for IC50. What if you use not IC50 but other metrics, e.g. IC90? IC10? What if you ignore prevalence of strains or just take Ab titer against one (perhaps dominant) strain? Will such a model fail to fit the data well?

Using data from another cohort to test model predictions is a fantastic way to determine if the model works. However, I am not sure if the metric used for that (correlation coefficient) is the right one. I think one needs to do a better regression analysis - after looking at Fig 7 - the relationship may be nonlinear - that's weird. Why? Can you show that alternative models have a lower predictive power when applied to the another cohort?

Minor comments

When you send the paper, make sure that all references are done properly. I see a lot of "error" in referencing (line 450, 835)

The panels that show correlation between different parameters that have the same range, use square dimensions for the plot, not the golden ratio.

When mentioning lower AIC values, put the Delta value in. A difference of 1-2 in AIC is not considered to be impressive, so just stating this without providing numbers is poor science.

nAb titers should be shown in units.

Fig S1 - why is the relationship between these two parameters nonlinear? Needs an explanation.

Reviewer #3

(Remarks to the Author)

My concerns have been addressed in the revisions.

Responses to reviewer comments

Reviewer #1 (Remarks to the Author):

Edge *et al.* use data from the PROVENT trial to predict the neutralising antibody (Ab) titre from the serum concentration of the monoclonal antibody (mAb) Evusheld and predict the protective effectiveness of the mAb and the duration of protection against different SARS-CoV-2 variants. The main results are a prevalence-adjusted and fold- IC_{50} titre of 458 corresponding to at least 50% protective efficacy (~65 IU/ml) and predictions of the duration of protection above 50% against different variants (and in the presence of multiple variants by prevalence-adjusted titres).

The main strength of this work is the use of individual-level serum concentration data. The manuscript is well written and clear, and the methodology is appropriate (besides concerns about the suitability of the model, see comment 4). However, the results seem to be lacking sufficient relevance and novelty for a journal like *Nature Communications*, a more field-specific journal may be more appropriate. Moreover, the results (model predictions) are not validated.

Major comments:

Comment 1. Lacking relevance:

Monoclonal antibodies are largely no longer used to treat COVID-19. Instead, antivirals have been shown to be much more robust to the emergence of new variants and are highly effective. Here, the authors present an analysis of one trial of one particular mAb (that is no longer used to treat COVID-19), against variants that are no longer circulating. Also, although COVID-19 treatments are primarily for the immunocompromised, their data is not clearly related to such populations (as the PROVENT trial also includes participants with increased risk of exposure, line 132). It is unclear that there is a use case for the current model proposed by the authors.

Author response:

We thank Reviewer 1 for their comprehensive review of our manuscript.

While antivirals are effective for the treatment of COVID-19, we believe that mAbs are still an important tool in COVID-19 prevention, particularly for the most vulnerable, such as immunocompromised populations where severe infections are more likely. We note that sipavibart has received approvals in the EU¹ and Japan.²

The genetic variability of SARS-CoV-2 presents a significant threat to the efficacy of preventative interventions. Although the model from PROVENT was fit against variants that are no longer circulating, this presented a rare opportunity to model efficacy as the trial spanned both a period where efficacy was observed against alpha and delta, and where it was lost following the emergence of omicron.

Furthermore, the aim of this work was to provide a variant-agnostic association between nAb titres and efficacy. It was hypothesised that through prevalence-adjustment the associated nAb titre–efficacy relationship is independent of the SARS-CoV-2 variant and the population. This assumption is now externally validated through the inclusion of data from the SUPERNOVA clinical trial in immunocompromised participants³ [see the Results, starting at line 378]. We believe that the concordance demonstrated between model predictions and observed data for a wide range of SARS-CoV-2 variants, as well as similarities in the parameter estimates from the PROVENT-only and pooled ToP Cox models, provide reassurance that the nAb titre–efficacy relationship is generalisable. Therefore, there is a clear case for the utility of the model.

The mutation rate of SARS-CoV-2 highlights the urgent need for understanding the efficacy of novel products against the current variant landscape with expedited clinical trials. The approach described in this paper enables both faster development of mAbs using a surrogate nAb-titre endpoint to link to efficacy, and rapid assessment of existing mAbs against emerging variants using only pharmacokinetics (serum mAb concentration) and the variant-specific in vitro half-maximal inhibitory concentration (IC₅₀) value.

Comment 2. Lacking novelty:

Serum concentrations of mAbs have previously been related to neutralising titres.⁴ Inclusion of different variants in the same model by scaling nAb titres has been reported previously^{5, 6} and the 'variant transportability assumption' (lines 478-479) of correlates of protection has been investigated for early variants and SARS-CoV-2 vaccination.⁵ The authors have previously published a quantitative relationship between mAb concentration and protection.⁷ A relationship between protective effectiveness of mAbs and the nAb titre has also been shown previously and used to predict the duration of protection, with the inclusion of variations the neutralising potency (IC₅₀) against different variants.⁶ As the authors state, pemivibart received emergency use authorization based on an immunobridging strategy (lines 82-83), so how does their work add to existing and already applied immunobridging strategies for mAbs?

How does this work compare to existing approaches, what is the novelty, and what is the added benefit compared to existing methods? The research context is not sufficiently considered and discussed in the manuscript. In this reviewer's opinion, after such a discussion of the existing literature it is difficult to see particular novelty in the current study.

Author response:

The publication by Clegg LE *et al.*⁴ was developed by AstraZeneca as a precursor to the work presented here by confirming the ability to predict nAb titres using in vitro IC₅₀ values and serum mAb concentrations. This is used as a tool to estimate the participant-level daily nAb-titre profiles in this manuscript. Building on that previous work, the present analysis establishes a method to quantify and externally validate (using SUPERNOVA) the relationship between predicted nAb titres and efficacy, whilst controlling for the variant landscape. Therefore, we believe that these two pieces of work should be considered as sequential steps to a common goal.

Stadler's ground-breaking paper explored the association between antibody dose and efficacy of therapy using log-binomial regression models.⁶ While this work represented an important first step, it was conducted using aggregated study data and did not account for study-level intricacies such as the subject censoring and

differences in follow-up. Stadler *et al.* provided a quantitative means to predict the therapeutic efficacy of mAbs against specific variants under different dosing regimens; however, they did not attempt to combine their data against multiple variants or adjust for changing viral prevalence. Therefore, we believe that the work presented in our paper is novel, building on the previous articles exploring the relationship between nAb levels and clinical efficacy. Our study also captures variability in viral prevalence during the trials and the data presented here highlights the significance of adjusting for changing viral prevalence, for example, the PROVENT trial spanned several dominant variants (some of which AZD7442 had reduced potency against).

The meta-analysis by Cromer *et al.* showed a link between population-level serum nAb titres and protection⁵; however, without individual-level data they were not able to quantify an individual-level correlate-of-momentaneous risk relationship. In addition, the work we present here is a significant improvement over Follmann *et al.*,⁷ as we include a substantially larger dataset and a refined modelling approach, as well as adjustment for the changing SARS-CoV-2 variant landscape at the individual-participant level whereas, in contrast, Follmann *et al.* was for a trial with a single circulating genotype. The inclusion of a second individual-level test dataset (SUPERNOVA) to validate predictions from the model makes our work notably original; to our knowledge, there is no comparable publication.

The immunobridge strategy for pemivibart was reliant on direct bridging to the observed historical efficacy estimate of 71% for adintrevimab against the Delta variant. Despite the fact that pemivibart failed this immunobridging hypothesis, an Emergency Use Authorization was still granted. Whilst not directly stated, this hints that the FDA recognised the value of a mAb targeting COVID-19, albeit with slightly lower efficacy than that observed for adintrevimab during the earlier part of the pandemic. This highlights several challenges with existing immunobridging strategies. Firstly, the binary null hypothesis is strictly linked to the observed efficacy in the historic trial without flexibility. Therefore, the ability of our current work to flexibly identify nAb-titre targets associated with different levels of efficacy represents a significant advance. Secondly, direct bridging must be done by variant and, therefore, fails to characterise protection against the complex variant landscape we

now find ourselves in. Our work accounts for this by adjusting for both variant prevalence and alternate variant neutralisation when estimating efficacy. We have included the above points in the introduction, lines 82 to 98, and 112 to 123.

Comment 3. Lack of validation of results:

The authors make a problematic statement as to the validation of their model prediction by stating that their results are “in agreement with observed data from the PROVENT final analysis” (l. 290-291) which is not surprising as this was the data used to calibrate the model. Since there are already existing works doing a similar analysis to that reported here (see above), it seems external validation might be possible, but the authors do not compare their model with existing models relating protective effectiveness of mAbs to Ab titres (e.g., [4] and [3]). The authors openly admit the need for validation in the discussion, but this does not in any way reduce the need to validate results.

Author response:

We agree that this was a weakness of the previous submission and had intended to publish a validation as soon as SUPERNOVA data became available to us. Fortunately, this is now possible and we have included these data in the manuscript [see the Results, starting at line 363].

Comment 4. Model choice:

The authors use a model that predicts negative effectiveness for low nAb titres which is biologically implausible (l. 299). They compare this model with another model that predicts high efficacy at low titres that “may not be biologically feasible” (Supplement, l. 146). Model predictions of the threshold of protection differ greatly (458 vs 160) which reduces the confidence in either estimate. Why are the authors not considering another model that is biologically plausible with non-negative effectiveness and allowing for a steeper increase in effectiveness to fit low effectiveness at low titres and higher effectiveness at higher titres? Generally, the conclusions of a study should not be so dependent on modelling assumptions that are clearly implausible.

Author response:

We detail three models in the manuscript [see the Supplementary Appendix, section 5, starting at line 151, including Supplementary Table S1; Supplementary Fig. S4; Supplementary Fig. S5]. The first is a single-parameter model that is supported by the biologically plausible assumption that the nAb-efficacy curve passes through the origin. The second two-parameter model relaxes this assumption and allows more flexibility, so that the curve is data driven. The third also relaxes this assumption whilst adjusting for baseline covariates (region, sex and BMI), all of which were statistically significant in the model.

The one- and two-parameter models from the ToP model built from PROVENT data were assessed in terms of their goodness of fit externally comparing model-predicted estimates of efficacy to those independently observed in SUPERNOVA. This was done separately for instantaneous and average overall efficacy [see the Methods lines 274–293]. The absolute mean difference between predicted and observed efficacy are 13.4% following the one-parameter approach compared with 8.7% following the two-parameter approach when evaluating the overall instantaneous efficacy, and 12.6% following the one-parameter approach compared with 5.0% following the two-parameter approach when evaluating the overall average efficacy. Not only did the one-parameter model have lower predictive accuracy, it also over-estimates efficacy and is therefore considered an anti-conservative approach. A further complication of restricting the model based on the assumption of zero efficacy at zero nAb titres is that the confidence intervals are artificially tight because a common variance is then applied at all levels due to the single parameter (Supplementary Fig. S5).

The two-parameter model was then compared to a similar model accounting for baseline covariates. Although a lower Akaike information criterion was achieved when adjusting for baseline covariates (indicating a better model fit), there were only small changes in the nAb titre-efficacy relationship (Supplementary Fig. S5), and thus the more parsimonious model was chosen to allow for more generalisable applications.

We did explore the Emax model for PROVENT (not included in the manuscript), but due to a long period of no placebo cases this returned a step function around the level of nAb titres at the start of this period, which is not biologically credible as changes in efficacy are expected to be incremental with level of nAb titres. On balance, we felt that although the two-parameter model does give some spurious results at very low titres this will not affect applicability of the model, and predicted efficacy from this model better aligns with external data from another mAb clinical trial (SUPERNOVA).

Reviewer #2 (Remarks to the Author):

Positive feedback. The text is short and to the point. Easy to read. Rationale and most details are well explained and straightforward to follow.

Major comments

Comment 1:

The authors make several assumptions to arrive to the conclusion that "a nAb titre of 458 (~65 IU/ml) corresponded to $\geq 50\%$ efficacy". Is this conclusion robust? Per my reading of the paper, I do not see analyses that illustrate robustness (and translation potential to other settings/studies). Critical assumptions seem to be the following:

- 1) nAb decay is the same in all individuals -> but obviously it is not -> if such variability is introduced how is the number 458 changed?
- 2) authors use \log_{10} Ab as predictor. This seems weird, it should be Ab titre as mechanistically Abs and not log of Abs should be predictors of protection.
- 3) Authors normalize Ab titre by IC50 from a specific assay. Why IC50? Would the model without normalization fail to fit the data? What about IC10? IC90? IC99?
- 4) Ab titres were weighted using log titres. Why not linear?

Author response:

We thank Reviewer 2 for taking the time to review our manuscript and we hope that our responses below adequately address their concerns.

We agree that the manuscript would benefit from analyses that illustrate the robustness of our findings and have therefore included additional data from the SUPERNOVA clinical trial of sipavibart to externally validate the model proposed. The SUPERNOVA study took place during a different SARS-CoV-2 variant landscape, tested a different mAb than the combination tested in PROVENT, and was conducted in an immunocompromised population, so we believe that this also provides some evidence of generalisability [see the Results, starting at line 363].

In our analysis, we use individual participant predictions of serum mAb concentrations over time, derived from population pharmacokinetic (popPK) models for AZD7442 and sipavibart, to predict daily nAb titres for each individual. These popPK models are previously published and give individual predictions for the decline in mAb levels over time, fit to each participant's measured serum mAb concentrations.⁴ Therefore, this approach does indeed capture participant-specific variability in mAb decline over time. Additionally, the prevalence adjustment reflects changes in nAb levels (derived as serum mAb concentration divided by the prevalence-weighted IC_{50}) to different variant combinations circulating over time in each participant's region. In this sense, individual variability is introduced into the model here. On revision we have included additional analysis of a model that includes baseline covariates, but the results of this are very similar to the more parsimonious model [see the revised Supplementary Appendix, section 5].

Given the log-normal distribution of nAb titres as shown in Supplementary Fig. S6, it is appropriate to analyse the relationship between log-transformed nAb titres and efficacy.

As described (**E 3**) in the revised Supplementary Appendix [line 128], IC_{50} values are used to convert serum mAb concentrations to a "standardised" or "prevalence-adjusted" nAb titre according to variant prevalence (proportion of variants in a specific region of the world corresponding to a given participant). This is necessary as variants change rapidly, prevalence is not globally uniform, and mAbs can have

significantly different potencies against different variants. The approach taken here to predict nAb titres using mAb potency (in vitro IC₅₀ values) and measured serum concentrations follows from the methods of Clegg LE *et al.*⁴ and have shown to be predictive of clinically measured neutralising titres. The choice of IC₅₀, as opposed to IC₁₀ or IC₉₀, is aligned to the available clinical nAb data (observed nAb titres) being reported as a 50% neutralising titre and previous confirmation that such prediction is in agreement with measured or observed nAb titres from clinical samples.⁸

Additionally, using IC₅₀ as opposed to IC₈₀ or IC₉₀ reduces the impact of the slope of the neutralisation curve that could vary between participants or between SARS-CoV-2 variants. It is also important for the source of mAb potency information to be consistent across all variants, i.e., measured in the same assay method. In this work we are utilising a lentiviral-based pseudovirus assay where all assay parameters (viral input, incubation time, etc.) are consistent across all SARS-CoV-2 variants tested. Thus, we can compare or combine titres across different variants or utilise the same conversion from titre value to IU/mL.

Comment 2:

Data shown in Suppl Fig 3 (probably should be in main text as these are original data) are interesting but it is hard to understand why treated individuals accumulate infections at much faster rate after day 156. This may indicate biases in the data. Also, the rate of accumulation of cases in treated cases does not speed up but is expected due to individuals losing protection at different rates as Abs decay differently in different individuals. This contradiction must be explored.

Author response:

We have chosen to remove the original Supplementary Fig. S3 from the manuscript as we are now using SUPERNOVA data to validate the PROVENT ToP Cox model. Supplementary Fig. S3 was originally included in the Supplementary Appendix to give context around our choice of changepoints in the piecewise linear model. It is somewhat difficult to interpret outside of this context because the x-axis is log(time). The data are a little unusual, likely because of significant behavioural changes during the COVID-19 pandemic (for example, the licensure of a vaccine was associated

with unblinding/drop out in this study). Interpretation of event rates over time is made harder because of seasonality and the 2:1 randomisation. In the revised document, Supplementary Fig. S2 (PROVENT) and Supplementary Fig. S3 (SUPERNOVA) may be easier to interpret [see associated text in the Results from line 334]. The change in rate of events during PROVENT is driven, in part, by the change in global viral prevalence or pandemic intensity, change in SARS-CoV-2 variants in circulation (to Omicron variants, which AZD7442 had reduced potency against), along with the decline in serum mAb concentrations over time. Thus, this is not a contradiction as such, but something we expect and have accounted for in the model.

Comment 3:

The key element of many studies of Abs and protection against COVID19 include some form of normalization of Ab titres, including those against the variants (e.g., 37507368). The idea seems reasonable but I have not seen much analysis that shows that such normalization is important, i.e., alternative models fail to fit the data. Another big concern is how to translate predicted numbers (e.g., 458) to other studies who do not use proprietary systems to measure nAb titres. This must be addressed.

Author response:

We chose to adjust nAb titres according to the variant landscape because during the clinical trials several variants with different mAb potencies emerged and became dominant. Differences in potency directly impact the mAb concentration required to be effective, for example, AZD7442 is much less potent against several Omicron variants (e.g., BA.1.1 has an IC_{50} of 466) compared to Wuhan (IC_{50} of 2.2), hence a higher mAb concentration was required to maintain the same level of neutralising activity. Not accounting for this is expected to lead to bias. We explored a model relating efficacy directly to serum mAb concentration using data from PROVENT. It was observed that the model overestimated efficacy compared to those observed from the SUPERNOVA data. Other work has dealt with this by splitting the risk period (for example into pre- and post-Omicron periods).⁹ However, this approach is primarily applicable to the early days of the pandemic which saw serial dominance of specific variants as opposed to the complex variant landscape we now find ourselves

in. Our work is strengthened by adjustment of individual nAb titres according to mAb in vitro IC₅₀ values and leads to a variant agnostic understanding of the relationship between nAb titres and protection.

Moreover, normalising to variant prevalence allows us to directly account for changes in the variant landscape on a regional level and assess the true nAb titre-efficacy relationship independent of the SARS-CoV-2 variant. This allows us to compare antibody responses across different studies and populations, reducing variability introduced by differing variant distributions. Any model that failed to account for prevalence would be associating efficacy to neutralisation against variants that may not be in circulation, which would likely result poorer estimation. Accounting for variant-specific prevalence not only provides a more robust perspective into the association between serum concentration and efficacy but is necessary to avoid spurious associations.

We agree that to compare our findings to other published data, the nAb titres should be expressed in international units (IU/mL). Table 2 has been added to show the nAb titres in IU/mL corresponding to specified levels of efficacy. Because in our work mAb potency is measured in the same pseudovirus assay method with consistent assay parameters across all SARS-CoV-2 variants, it justifies combining the titres to different variants based on prevalence and also utilisation of the same conversion from titre value to IU/mL. The conversion factor was derived by assessment of First WHO International Standard NIBSC 20/136 that was established using sera potent against ancestral SARS-CoV-2 Wuhan variant and has been accepted in the field. However, because the parameters of the pseudovirus assay are the same across all variants (the only change in the assay is the sequence of the SARS-CoV-2 spike protein), the same conversion factor can be applied to all variants. Further development of broadly neutralising international standards with accepted IU/mL can further facilitate comparison of the titres across different neutralisation methods by different laboratories.

Comment 4:

I find it unacceptable that the data from the trials are not freely available but require requests to the company. The company should be able to anonymize the data and make them available as a requirement for publication.

Author response:

The trial data can be made available to qualified researchers on request via the Vivli platform, as is standard for AstraZeneca. Certain requirements need to be met, as detailed in the data sharing statement.

Comment 5:

Methodology of the modeling must be moved to main text with ALL equations numbered.

Author response:

All equations are now numbered in the revised Supplementary Appendix. We believe that retaining equations in the Supplementary Appendix will make the main text more accessible for a non-statistical audience. Also, as some equations are applied multiple times under different methods described in the Supplementary Appendix, keeping all equations in this document enables the reader to easily cross-reference. This further aids with the flow of the article.

Comment 6:

The model with intercept is useful but that it gives negative efficacy raises concern. Why not use a model that does not allow for negative values but does have intercept. That would be a deviation from standard hazard models but authors could devise a method based on likelihood to use a user-defined model to fit the model to data.

Author response:

This comment was also raised by Reviewer 1. We concluded that the only biologically plausible assumption to make for the model is one of zero efficacy at zero nAb titres (any other intercept, negative or positive, does not align with our understanding that mAbs confer protection via neutralising antibody titres only). This assumption is explored in the single-parameter model, which is why we included the discussion on this in the manuscript [see the Supplementary Appendix, section 5, starting at line 151, including Supplementary Table S1; Supplementary Fig. S4; Supplementary Fig. S5].

Minor comments**Comment 7:**

Figures need improvements. For multi-panel figures, each panel should be labeled as A/B/C (or similar), e.g., Figure 1.

Author response:

Agreed, we have made this change throughout.

Comment 8:

It is impossible to see different line styles in Figure 1. Perhaps make a multipanel figure? Or use log-scale in A? Fig 1B is also impossible to follow -> perhaps making figure larger in y axis could help.

Author response:

This figure has been replaced, please see new Fig. 2.

Comment 9:

I do not understand Figure 2 -> why are there only 3 points for titres? Is the model fit here good? The fit does not seem good to me, averages are not explained well.

Author response:

We agree, this was not the best way of describing model fit. We have removed the three points and replaced them with an external assessment of instantaneous efficacy from the PROVENT ToP Cox model using SUPERNOVA data, please see updated Fig. 3.

Comment 10:

Are Figs 3&4 that useful for main text? Fig 4 is assumption, and Fig 3 is hard to interpret. Can you provide statistics to show that two models provide similar predictions besides a visual one?

Author response:

We have replaced these figures. In Fig.7 Lin's concordance correlation coefficient is now used to summarise the agreement compared with study results over multiple variant-driven endpoints from SUPERNOVA³ against those predicted by the model, which assesses the generalisability to assess efficacy against other variants.

Comment 11:

Fig S1 -> what are the relationships here? Authors should fit the hill-based relationship and report hill coefficient -?> it seems that for some variants, relationship is not linear (e.g., BA.2) -> Why?

Author response:

Supplementary Fig. S1 presents the plot of observed versus predicted nAb titres to assess concordance. Predicted nAb titres were derived as the observed serum concentration (ng/mL)/in vitro IC₅₀ values (ng/mL), as described in **E 1** [see

Supplementary Appendix, line 33], where the in vitro IC₅₀ values are those presented in Table 1. This relationship assumes a Hill coefficient of 1.

From Supplementary Fig. S1 we agree that allowing the Hill coefficient to vary here for BA.2 may lead to an improvement in concordance for this variant, although generally the concordance is still good (Lin's concordance correlation coefficient: 0.68; 95% confidence interval: 0.64–0.72). The difficulty with allowing the Hill coefficient to change when predicting nAb titres is that estimating the variant-specific slope would necessitate collection of observed nAb titres for all variants for which predictions are to be made.⁴ Hence, instead of measuring mAb potency in vitro and utilising already collected serum mAb concentration patient data to derive predicted titres, this would require large scale collection and testing of patient serum samples to generate new observed nAb titres, which takes months and would erode the ability to deploy this model for rapid decision making. The purpose of the analysis here is to expediate clinical evaluation of mAbs, which is no longer possible once the Hill coefficient is allowed to vary. Therefore, the assumption of a Hill coefficient of 1 is a practical compromise when deriving predicted nAb titres as it still provides adequate concordance. We have added a section on this to the Discussion [lines 408–418].

Comment 12:

The extended Cox model (page 7 in supplement) was used -> is this a good model? Which alternative models did you try and why other models failed?

Author response:

We chose to use an extended Cox model as this handles time-to-event data well along with time-varying variables and is familiar to most statisticians working in the industry. As previously noted, we tried to fit an Emax model but found that it did not converge sensibly. Alternative parametrisation and transformations of the prevalence-adjusted nAb titres are explored in the revised Supplementary Appendix. Model-derived estimates are also externally validated against alternative methods, such as kernel-smoothed hazards (Fig. 4) and Poisson regression with robust variance (Fig. 5 and Fig. 6).

Comment 13:

Fig S3 -> I do not understand log(t) value of 4. Please use log-scale but numerics that could be understood by humans.

Author response:

Previous Supplementary Fig. S3 has been removed given the model is now externally validated using SUPERNOVA data.

Comment 14:

I did not see a model for nAb dynamics. It should be listed along with estimated parameters. Did authors use nonlinear mixed effects to predict nAb kinetics?

Author response:

As described in the revised Supplementary Appendix [line 17 onwards], nAb titres were derived by dividing the daily serum mAb concentrations predicted by the population pharmacokinetic models for AZD7442 and sipavibart by the prevalence-adjusted IC_{50} . Prevalence-adjusted IC_{50} was derived by combining in vitro potency (IC_{50} values) with SARS-CoV-2 variant surveillance data that indicated the proportion of variants in the region of the world corresponding to the patient location. The popPK models referenced in the manuscript allow for individual-level variation with a nonlinear mixed-effects model.¹⁰

Reviewer #3 (Remarks to the Author):

Edge *et al.* describe the development of a model to determine a neutralising antibody (nAb) threshold of protection for monoclonal antibodies targeting SARS-CoV-2 in the context of pre-exposure prophylaxis against COVID-19. Using data from the PROVENT clinical trial of the mAb combination AZD7442, the authors determined that a nAb titre of 458 (~65 IU/mL) corresponded to >50% efficacy against

symptomatic COVID-19. This model, based on predicted nAb titres and variant prevalence, was validated using data from multiple SARS-CoV-2 variants. The authors' conclusions are well-supported and offer valuable insights into the neutralising antibody levels required for protection against COVID-19, potentially aiding regulatory approval of SARS-CoV-2 mAbs based on surrogate endpoints and reducing the need for large efficacy trials for new variants. However, the manuscript would benefit from additional data or discussion, as outlined in my comments below.

Comment 1:

Line 34. "This study identified a threshold value for neutralising antibody titres (nAb) associated with clinically relevant protection against symptomatic COVID-19 for vulnerable populations". This statement is a bit misleading, as it suggests that the trial population was comprised of immunocompromised individuals which was not exclusively the case.

Author response:

We thank Reviewer 3 for their detailed review of our manuscript.

We have added additional data collected from immunocompromised individuals to externally validate the model, so we suggest that this language is kept.

Comment 2:

Were there enough cases in the clinical study to determine a threshold nAb titre for protection against severe disease or asymptomatic infections?

Author response:

In the analysis presented here, we consider the primary endpoint for the clinical trial; we have much more data at this endpoint. However, it would be appropriate for future work to consider both of these endpoints if pooled data provides a suitable number of events.

Comment 3:

Did the authors measure mucosal antibody in nasal or oral swabs to understand levels of mucosal Abs required for protection against symptomatic disease? If not, the authors should comment that levels of nAb in the upper respiratory tract are likely responsible for the observed protection which will only be a small fraction of the serum Ab level given low efficiency of serum Ab transudation. This is probably the reason why relatively high concentrations of serum nAb are required to protect against upper respiratory disease.

Author response:

While nasal lining fluid (NLF) concentrations of sipavibart are not yet available, concentrations of tixagevimab-cilgavimab (AZD7442) in NLF are available from the first-in-human study in healthy volunteers.¹¹

The level of partitioning of AZD7442 from serum into NLF was used, along with a theoretical target of 80% viral neutralisation in the NLF (and an assumption of a Hill coefficient of 1), to inform dose selection, and dose modification in response to emerging variants, for tixagevimab-cilgavimab. This has been previously described.³
¹⁰ The limitation of this approach to set a target serum concentration for a mAb is its reliance on these underlying unvalidated assumptions, along with the additional cost and burden to patients to collect NLF samples in clinical trials. To avoid these limitations and assumptions, this analysis, analogous to many exposure-response analyses, seeks to use the available clinical trial data to relate efficacy to easily measured quantities: serum mAb concentrations and in vitro IC₅₀ values. Additional text to this effect has been added to the last paragraph of the revised Discussion [line 491 onwards].

Comment 4:

The study population was SARS-CoV-2 naïve, which differs from the immune status of the majority of the global population today. Can the authors comment on how the nAb COP curve might shift in the setting of pre-existing immunity? For example, is it

possible that T cell immunity might lower the required nAb threshold associated with 50% protection?

Author response:

The model seems robust when applied to SUPERNOVA, which is a trial conducted in the setting of pre-existing immunity, albeit with an immunocompromised population. Future development of mAbs is likely to focus on immunocompromised individuals, and this model seems to be generalisable to this population.

Comment 5:

It would be helpful to also indicate nAb titres in IU/mL in figure 2 and to indicate the titre require for 70 or 80% protection which is the more likely target for future monoclonal therapies.

Author response:

We thank the reviewer for their feedback and have added Table 2, which presents the efficacy level for 40, 50, 60 and 70% efficacy along with the conversions to IU/mL. Previously, we focused on 50% protection as at the time this aligned with FDA guidance for COVID-19 vaccines. The FDA guidance has since been rescinded. Rather than propose a threshold ourselves, we believe this needs to be agreed in advance with the applicable regulatory body. Following this regulatory discussion in SUPERNOVA, the go criteria was agreed under the principal that 40% was a minimum clinically relevant target for immunocompromised populations.³ Whilst highlighting the 40% efficacy level, the paper now focuses on the overall external validation of the ToP model, from which a target efficacy can be flexibly defined.

Comment 6:

The raw data used for the COP modeling should be provided as a supplementary excel sheet.

Author response:

Raw data files would be large and in a format unlikely to be meaningful to the public. The trial data can be made available to qualified researchers on request via the Vivli platform, as is standard for AstraZeneca. Certain requirements need to be met, as detailed in the data sharing statement.

References

1. European Medicines Agency. Kavigale approval (January 20 2025). https://www.ema.europa.eu/en/documents/overview/kavigale-epar-medicine-overview_en.pdf (
2. AstraZeneca. AstraZeneca's long-acting monoclonal antibody Kabigale Approved for the Treatment of SARS-CoV-2-Induced Infections in Immunocompromised Patients. <https://www.astrazeneca.co.jp/media/press-releases/2024/2024122701.html> (2024).
3. Haidar, G. et al. Efficacy and safety of sipavibart for prevention of COVID-19 in individuals who are immunocompromised (SUPERNOVA): a randomised, controlled, double-blind, phase 3 trial. *Lancet Infect Dis*, [https://doi.org/10.1016/S1473-3099\(1024\)00804-00801](https://doi.org/10.1016/S1473-3099(1024)00804-00801) (2025).
4. Clegg, L. E. et al. Serum AZD7442 (tixagevimab-cilgavimab) concentrations and *in vitro* IC₅₀ values predict SARS-CoV-2 neutralising antibody titres. *Clin Transl Immunology* **13**, e1517 (2024).
5. Cromer, D. et al. Predicting vaccine effectiveness against severe COVID-19 over time and against variants: a meta-analysis. *Nat Commun* **14**, 1633 (2023).
6. Stadler, E. et al. Monoclonal antibody levels and protection from COVID-19. *Nat Commun* **14**, 4545 (2023).
7. Follmann, D. et al. Examining protective effects of SARS-CoV-2 neutralizing antibodies after vaccination or monoclonal antibody administration. *Nat Commun* **14**, 3605 (2023).
8. Clegg, L. E. et al. Pharmacokinetics and Safety of the SARS-CoV-2 Monoclonal Antibody Sipavibart are Consistent With Those of Tixagevimab/Cilgavimab. In: *ECCMID* (2024).
9. Seekircher, L. et al. Anti-Spike IgG antibodies as correlates of protection against SARS-CoV-2 infection in the pre-Omicron and Omicron era. *J Med Virol* **96**, e29839 (2024).
10. Clegg, L. E. et al. Accelerating therapeutics development during a pandemic: population pharmacokinetics of the long-acting antibody combination AZD7442 (tixagevimab/cilgavimab) in the prophylaxis and treatment of COVID-19. *Antimicrob Agents Chemother* **68**, e0158723 (2024).
11. Forte-Soto, P. et al. Safety, tolerability and pharmacokinetics of half-life extended severe acute respiratory syndrome coronavirus 2 neutralizing

monoclonal antibodies AZD7442 (tixagevimab-cilgavimab) in healthy adults. *J Infect Dis* **227**, 1153-1163 (2023).

Responses to reviewer comments

Reviewer #1 (Remarks to the Author):

In their revised manuscript Edge et al. include additional data from the SUPERNOVA study to validate their model. The main strengths of this manuscript are the use of individual-level serum concentration and infection data from two separate studies to establish and validate a correlate of protection for monoclonal antibody prophylaxis.

My previous comments concerned the lack of relevance, novelty, validation of results, and the model choice. In their updated manuscript, Edge et al. have added validation of their results by adding data from a second study and better explained the relevance and research context and highlighted that the main novelty lies in the use of individual data and the use of the prevalence-weighted mean IC₅₀s to account for multiple variants circulating simultaneously.

The authors have sufficiently addressed my previous comments and I have only some minor comments on the updated version of the manuscript.

Minor comments

Comment 1:

L. 223: for the SUPERNOVA study, some IC₅₀s were imputed at the upper assay limit of 1,000 ng/mL. Does this imputation affect the model estimates, i.e., if they were imputed at, e.g., 1,500 or 2,000 ng/mL, would that affect the conclusions and the comparison with the PROVENT ToP model?

Author response:

The appropriateness of the imputed value of 1000 ng/mL used for variants with an F456L mutation can be assessed from looking at how well the model externally predicts the efficacy against these variants in Fig. 6. The value of 1,000 is the ULOQ of the assay that is utilised for measuring the IC₅₀ values. For both 3 and 6 months post any dose, this selected value underestimates the observed efficacy by 14.2 and 27.6, respectively. Therefore, imputation at any larger value would only worsen the model performance, which is already conservative (see below figure). This point has been added to the discussion section at lines 272–274.

Comment 2:

L. 246: The authors state that their method is “independent of baseline nAb titers” but provide no evidence or references for this statement. In the PROVENT study the control group’s titers were imputed to be zero and in the SUPERNOVA data comparator titers appear to also be negligibly low in the comparator group. However, e.g. the RECOVERY study showed different effectiveness of mAbs in seropositive and seronegative groups [1] and not all immunosuppressed patients have negligibly low titers. Would the method hold for an immunosuppressed cohort with low but non-zero titers?

Author response:

For the PROVENT analysis, this was conducted in an unexposed and unvaccinated population with a while-on-treatment policy applied to censor intercurrent events expected to increase an individual’s nAb titres (e.g. such as vaccinations or receipt of another mAb). MAb effectiveness in this population was therefore not expected to be confounded by baseline nAb titres. This justified the appropriateness of calculating predicted nAb titres from serum mAb concentration and IC₅₀. Additionally, by excluding any baseline nAb titres, this focuses the analysis specifically on nAb titres provided by the prophylactic mAb.

As observed in the RECOVERY study, mAb efficacy is expected to be positively correlated with the level of baseline nAb titres. This motivates the choice of

developing the PROVENT ToP model based on predicted nAb titres as a conservative approach to optimise patient outcomes in the presence of baseline nAb titres (clarified in lines 197–200). From an application perspective, this conservative approach is supporting in ensuring that all participants dosed would be expected to achieve nAb titres, and associated efficacy, in the target range, regardless of their baseline nAb titre against a given SARS-CoV-2 variant.

To further assess whether this approach would result in useful predictions in a population with higher and more variable baseline nAb titres, the predictiveness of the PROVENT ToP model was assessed against data from SUPERNOVA.

SUPERNOVA was conducted in an immunocompromised population excluding participants who received a SARS-CoV-2 mAb 6 months prior or received a COVID-19 vaccine or had COVID-19 3 months prior. In this population, baseline nAb titre levels were expected to vary across both individuals and SARS-CoV-2 variants based on exposure and vaccination history. Figures 4–7 all demonstrate good agreement with PROVENT ToP model predictions and observed efficacies from SUPERNOVA, including across a range of variant-specific mAb efficacies (Figures 6–7), where differing baseline nAb titre levels are expected. This supports that the modelling choice of using predicted titres results in reasonable predictions even without accounting for baseline nAb titres.

As the predicted nAb titres are included as an interaction with the binary treatment indicator, the choice of the imputed value used for placebo nAb titres will have zero effect on the predicted efficacy, as the hazard ratio is only modelled based on nAb titres from those receiving the mAb. This point is now clarified in lines 148–149 in the Supplementary Methods.

Comment 3:

The sentence in lines 482 to 484 was unclear to me.

Author response:

The author thanks reviewer 1 for their comment. This sentence has been revised in lines 267–268. The ToP model does not account for the model error in estimating the parameters from the popPK model (described in lines 368–396), which are used to derive daily serum mAb concentrations, and in turn the prevalence-adjusted predicted nAb ID₅₀ titres (lines 451–462). However, as described in lines 160–162, Fig. 5 shows in estimating the average overall efficacy over time, the confidence intervals from the PROVENT ToP model are conservative compared with those estimated from the observed data in SUPERNOVA.

Comment 4:

Fig. 7: the caption mentions CIs, but there were no CIs in the figure.

Author response:

In Fig 7 the abbreviation CI is used in the last column title in the top left table, referring to the bootstrapped confidence interval used for Lin's concordance correlation statistic.

References

1. RECOVERY Collaborative Group. Casirivimab and imdevimab in patients admitted to hospital with COVID-19 (RECOVERY): a randomised, controlled, open-label, platform trial. *The Lancet* 399, 665–676 (2022).

Reviewer #2 (Remarks to the Author):

Major comments

Comment 1:

The two models of how nAb titers relate to protection are apparently inadequate. Simpler model (no intercept) does not predict protection adequately (over-estimates) and two parameter model predicts negative protection at low nAb titers. Claiming that a better fit model is correct is false as it is biologically incorrect. You need to come up with the 3rd model that would be biologically plausible and predicts accurately protection levels. For example, a model with no intercept but with $n! = 1$. Alternative you could modify your 2 parameter model to predict 0 protection until $nAbs \leq A_{min}$, and increase after that. Unless you have a model that makes sense and fits the data well, the work is unfinished.

Author response:

The authors thanks reviewer 2 for the suggested model approaches. Based on the suggested model described at the end of comment 1, the following model was fit:

$$h(t_d; Z_i, X_i(t_d)) = h_0(t_d) \exp(\gamma_1 Z_i X_i(t_d))$$

where: $h_0(t)$ represents the baseline hazard function, Z_i is the treatment indicator (equals one for individuals randomised to tixagevimab–cilgavimab and zero otherwise),

$$X_i(t_d) = \begin{cases} \log_{10}(\tilde{Y}_{i,t_d} - \delta + 1) & \text{if } \tilde{Y}_{i,t_d} > \delta \\ 0 & \text{otherwise} \end{cases}$$

\tilde{Y}_{i,t_d} is the prevalence-adjusted nAb ID₅₀ titre as described in (E3) (see Supplementary Methods), and δ defines a nAb ID₅₀ titre intercept, below which efficacy is zero. Optimisation of δ was first done using 5-fold cross validation, before the model was then refit on all PROVENT participants. The results of the optimisation algorithm applied to δ and the resulting threshold curve are displayed in Fig. S5, which is shown to possess the same properties described by reviewer 2. The threshold curve is also compared to the other model approaches considered in Fig. S6.

One limitation of this model is that the threshold slope is optimised based on individuals with prevalence-adjusted nAb ID₅₀ titres > 78.8. This leads to a reduction in event counts (118 to 84), resulting in wider confidence intervals.

Following the same approach taken in Fig. 7, the selected model was used to predict the variant-specific efficacies in SUPERNOVA. The concordance of these estimates vs the observed efficacies in SUPERNOVA are shown in Fig. S8, along with the comparison for the no intercept model described in (M2) (see Supplementary Methods). Although the new model has comparable concordance to (M2) for the period 3 months post any dose, considerably worse concordance is seen when compared over a period of 6 months post any dose. Both models result in worse concordance compared to the proposed model (M1) (see Supplementary Methods) when compared to Fig. 7.

A discussion of this model is given in lines 228–230, and 477–479 in the manuscript, with full details provided in lines 183–208 in the Supplementary Methods.

Comment 2:

You used methodology to predict nAb levels using combination of IC₅₀ and variant prevalence (E2). This is somewhat similar to what has been used by the Davenport group. But how do we know this expression works really better than anything else? You did not show any alternatives that do not explain the protection data. Again, per rigorous science is not enough to show what seem to work, it is critical to show what does not work. What is you don't use log for IC₅₀. What if you use not IC₅₀ but other metrics, e.g. IC₉₀? IC₁₀? What if you ignore prevalence of strains or just take Ab titer against one (perhaps dominant) strain? Will such a model fail to fit the data well?

Author response:

Based on the proposal from reviewer 2, a model with [based on (M1)] and without an efficacy intercept [based on (M2)], using the IC₅₀ of 2.2 ng/mL from the dominant Delta variant (as seen in Fig. S2) was fit. Note that the preceding period is dominated by Alpha, which has a similar IC₅₀ of 2.1 ng/mL. The results of this model are shown to have poor internal agreement with the observed trends in

instantaneous and average efficacy observed in PROVENT (Fig. S9), particularly during the period with lower observed efficacies due to the emergence of Omicron, which is associated with a larger IC_{50} (BA.1.1 466 ng/mL; BA.1 171.1 ng/mL). A model with the ability to account for fluctuations in mAb potency based on the current variant landscape is considered to have better utility against emerging variants as discussed in lines 384–401 in the Supplementary Methods.

Please note that only IC_{50} in vitro values were reported by Monogram Bioscience, so the author is unable to explore the appropriateness of IC_{10} and IC_{90} as suggested by the reviewer. The rationale for this choice was previously discussed in the previous round of review, and included below for ease of reference:

“The choice of IC_{50} , as opposed to IC_{10} or IC_{90} , is aligned to the available clinical nAb data (observed nAb titres) being reported as a 50% neutralising titre [see included figure in author response to comment 6] and previous confirmation that such prediction is in agreement with measured or observed nAb titres from clinical samples [2]. Additionally, using IC_{50} as opposed to IC_{80} or IC_{90} reduces the impact of the slope of the neutralisation curve that could vary between participants or between SARS-CoV-2 variants. It is also important for the source of mAb potency information to be consistent across all variants, i.e., measured in the same assay method. In this work, we are utilising a lentiviral-based pseudovirus assay where all assay parameters (viral input, incubation time, etc.) are consistent across all SARS-CoV-2 variants tested. Thus, we can compare or combine titres across different variants or utilise the same conversion from titre value to IU/mL.”

Comment 3:

Using data from another cohort to test model predictions is a fantastic way to determine if the model works. However, I am not sure if the metric used for that (correlation coefficient) is the right one. I think one needs to do a better regression analysis - after looking at Fig 7 - the relationship may be nonlinear - that's weird. Why? Can you show that alternative models have a lower predictive power when applied to the another cohort?

Author response:

To clarify, Lin's concordance correlation coefficient (CCC) differs from a normal correlation coefficient as it measures the deviations against the line of perfect agreement $y=x$ [3]. On the other hand, typical correlation coefficients do not measure agreement. For example, the points on the line $y=2x$ would have a correlation of 1, but clearly this is a deviation from perfect agreement ($y=x$).

Based on Lin's CCC, in Fig. S8 the same evaluation used in Fig. 7 is applied to the model with no intercept and new model described in the response to the first comment from reviewer 2. The results show that these alternative approaches return lower predictive power.

Also note that Fig 7 is not the only method in which we assess the model predictions against the external cohort. In Fig. 4 and Fig. 5, the PROVENT ToP is used to predict the instantaneous and average trends in overall efficacy over time based on SUPERNOVA data. These are shown to have good alignment with the observed trends in these efficacy measures. Similarly, Fig. S4 shows the same comparison for the model with no intercept, showing again a lower predictive power for this alternative model.

Minor comments

Comment 4:

When you send the paper, make sure that all references are done properly. I see a lot of "error" in referencing (line 450, 835)

Author response:

The author thanks reviewer 2 for highlighting the referencing issues. These have been corrected in the new revision.

Comment 5:

The panels that show correlation between different parameters that have the same range, use square dimensions for the plot, not the golden ratio.

Author response:

The author thanks reviewer 2 for their comment and agrees that axes dimensions should be fixed to help assess the agreement. Fig. S4 (previously Fig. S1) has been updated based on the reviewer's feedback.

Comment 6:

When mentioning lower AIC values, put the Delta value in. A difference of 1-2 in AIC is not considered to be impressive, so just stating this without providing numbers is poor science. nAb titers should be shown in units. Fig S1 - why is the relationship between these two parameters nonlinear? Needs an explanation.

Author response:

Following reviewer 2's feedback, the author has referenced the magnitude in change in AIC in lines 142 and 192–193 in the Supplementary Methods.

Predicted nAb ID₅₀ titres are derived from serum concentration [ng/mL] divided by IC₅₀ [ng/mL] so are mathematically unitless. The observed/measured ID₅₀ titre is often presented as ID₅₀ titre without a unit shown [4,5,6]. The manuscript language has been updated to reflect this. Conceptually, ID₅₀ titre is a reciprocal sample dilution that inhibits the viral infection by 50%, i.e. if the sample dilution corresponding to 50% inhibition is 1/100, the ID₅₀ titre is 100. Because we determine the IC₅₀ (concentration of mAb that inhibits viral infection by 50%) and we know the concentration of that mAb in the serum sample from PK assessment, we can predict how much we need to dilute this sample to reach the IC₅₀ value corresponding to ID₅₀ titre or predicted titre in the same assay. This is illustrated in the schematic from Clegg et al., shown below.

Fig. S4 (previously Fig. S1) shows some deviations in the slope of the observed vs predicted nAb titres compared to the line $y = x$. This is indicative that the true Hill coefficient may deviate from 1 on a variant-by-variant basis. However, reasonable

concordance in the range of 0.6–0.7 was still obtained with this assumption. Although not perfect in modelling the data, applying a Hill slope of 1 is a practical compromise, as without this assumption the generation of observed nAb titre data and confirming the hill slope for each emergent variant would be required. This would reduce the ability to streamline the analyses by utilising the predicted nAb data (based on already collected clinical data combined with in vitro IC₅₀) instead of generation of observed nAb data for every new variant. This point is discussed in lines 200–207.

References

2. Clegg LE et al. Pharmacokinetics and Safety of the SARS-CoV-2 Monoclonal Antibody Sipavibart are Consistent With Those of Tixagevimab/Cilgavimab. ECCMID 2024. Poster LB026.
3. Lin L I-K. A Concordance Correlation Coefficient to Evaluate Reproducibility. *Biometrics* 45, 255–268 (1989).
4. Huang Y et al. Calibration of two validated SARs-CoV-2 pseudovirus neutralization assays for COVID-19 vaccine evaluation. *Scientific Reports* 11, 23921 (2021).
5. Anderson EJ et al. Safety and Immunogenicity of SARS-CoV-2 mRNA-1273 Vaccine in Older Adults. *N Engl J Med* 383, 2427–2438 (2020).
6. Ramasamy, Maheshi N et al. Safety and immunogenicity of ChAdOx1 nCoV-19 vaccine administered in a prime-boost regimen in young and old adults (COV002): a single-blind, randomised, controlled, phase 2/3 trial. *The Lancet* 396, 1979-1993 (2021).

Reviewer #3 (Remarks to the Author):

My concerns have been addressed in the revisions.